# Eco-Innovation Influencers: Unveiling the Role of Lean Management Principles Adoption

**João Leitão** [1],*, **Sónia de Brito** [2] **and Serena Cubico** [3]

1   Department of Management and Economics, Faculty of Social Sciences and Humanities, University of Beira Interior, Estrada do Sineiro, 6200-209 Covilhã, Portugal
2   Research Center in Business Sciences (NECE), University of Beira Interior, Rua Marquês d´Ávila e Bolama, 6201-001 Covilhã, Portugal; sonia.rosana.brito@ubi.pt
3   Department of Business Administration, University of Verona, I-37129 Verona, Italy; serena.cubico@univr.it
*   Correspondence: joao.correia.leitao@tecnico.ulisboa.pt; Tel.: +351-275-319-853

**Abstract:** This study analyzes the determinant factors of eco-innovation, considering business units with different levels of technological intensity (high technology versus low technology). It aims, in the first instance, to complement the approach on the determinants of eco-innovation in the existent literature by incorporating the novelty related to the analysis of the effects arising from the adoption of the lean management principles. Specifically, it aims to analyze the effects of the previously referred to determinant factors both on the economic performance and on the innovative performance of Portuguese industrial and service companies with different levels of technological intensity (high-tech versus low-tech). The conceptual model presented is of an innovative nature, since it includes four groups of determinant factors present in the literature, namely technology, market, public policies, and cooperation relationships, and adds a fifth group of determinant factors still to be explored empirically concerning the adoption of lean management principles. In the empirical approach, five research hypotheses arising from the literature review are tested, using secondary data collected from the Community Innovation Survey (CIS)—CIS 2010 for a total sample of 334 companies, made up of 95 high-tech companies and 239 low-tech companies. The conceptual model is tested using a logistic regression method, which indicated a suitable accuracy and reliability for the purposes of empirical tests. The empirical evidence confirms that most of the groups of determinants previously identified in the literature have a significant influence on eco-innovation. In addition, the empirical evidence obtained here indicates a positive and significant effect of lean management principles on eco-innovation.

**Keywords:** eco-innovation; community innovation survey; sustainable business models; lean management principles

## 1. Introduction

Sustainability is an extremely important topic that has been adopted by the business world, government policies, and society in general, as a way of safeguarding our ecosystem and a strategy to achieve competitive advantages. It is also considered a way to cope with economic, environmental, and social factors [1,2]. For a full sustainable performance of companies, not only economic factors but also stronger actions to promote the creation of jobs related to environmental activities are needed [3]. In economic terms, humans' increasing need to acquire new products leads to the excessive consumption of natural resources as well as an increased waste production. Therefore, one way to halt the reckless destruction of the planet is by promoting sustainable products and manufacturing processes [4–6].

Besides sustainability, innovation should emerge as a fundamental strategy to create an ecological society favouring the exploration of new business opportunities based on new, sustainable business models. In turn, given the growing globalization of economic activity and consumers' new needs, innovation has become an obligatory, wide-ranging challenge for companies [7,8], and as such, the more innovative a company, the more environmentally friendly it can become [9].

Factors such as technology and innovation are fundamental for the success of market strategies oriented towards sustainability [5]. Moreover, by adopting a sustainable environmental management, firms increase their competitiveness, reducing costs, improving quality, and applying innovative products and processes [10,11].

Competitiveness is, therefore, no longer exclusively a strategic question of differentiation, a leadership in costs, or a focus on one of these but is, above all, a question of innovation capacity [10]. It is in this context that innovation becomes an essential tool for firms' competitive survival, which should be (re)thought and planned, aiming to promote good, sustainable practices without jeopardizing the future society [2,5,12]. It is also worthwhile to mention that the role assumed by managers in the decision-making process, especially in the construction of collaborative partnerships open to sustainable innovation, assumes a fundamental importance [13].

Formulating a business model, developing a competitive strategy, marketing, and obtaining a substantial share of the market are some aspects in which innovation can and should be addressed [14].

Recently, eco-innovation has emerged as the result of integrating the philosophy of sustainability in the context of the business innovation process. This is a special type of innovation that reduces the negative impacts on the environment and is characterised by a double positive effect, producing new knowledge and improvements in environmental terms that allow the internalization of different externalities obtained in a given business context. Furthermore, it is through eco-innovation that new products, services, processes, technology, organisational structures, business models, institutions, practices, and social systems can be modified or introduced [15].

In this context, in recent years, eco-innovation has attracted growing interest among researchers who devote their efforts to identifying the determinants of eco-innovation (for example, in Reference [16]). Despite this, the literature on the determinant factors of eco-innovation still lacks further efforts and under different research lenses [17]. In addition, studies addressing the effects of lean principles on eco-innovation are almost nonexistent, with no evidence of how these principles behave, so as to positively or negatively influence the adoption of eco-innovative activities. Only a few of these principles have been explored in the literature on interrelated topics such as eco-innovation, project management, and resource-based view. Portillo-Tarragona et al. [18] show in their analysis that companies implement eco-innovation solutions at different levels of the value chain, designing and adopting advanced environmental management systems funded on the ISO 14001 and ISO 5000 standards. Still, in the field of lean, Pacheco et al. [19] presented a proposal for an integrated model focused on the systematic generation of eco-innovations.

In this line of reasoning and in order to address the gap identified, this study intends to contribute actively to the advancement of knowledge about the emerging topic related to the determinant factors of eco-innovation (see formulation of the research hypotheses: H1, H2, H3, and H4), and it also intends to analyse the possible effects of adopting lean principles on pro-eco-innovation orientation in order to improve the understanding of the extent to which the adoption of lean principles can act as a determinant of eco-innovation (as stated in the H5 hypothesis).

Consequently, it is considered that this study of the factors influencing the innovation process in the business context can contribute both academically and scientifically and to practice.

Academically and scientifically, by analysing the main determinant factors influencing firms' innovation processes, this research aims to increase knowledge about this recent topic and to contribute actively to advancing the knowledge of the emerging topic related to eco-innovation and sustainable business models. At the same time, it aims to make some progress in the study of innovation, in general, and the determinant factors of eco-innovation. With this, the overall objective of this study is, in the

first instance, to contribute to the literature of determinants of eco-innovation by unveiling the still unexplored role played by the adoption of lean management principles. In specific terms, it aims to provide an innovative application by assessing the differences between companies with different levels of technological intensity (high-tech versus low-tech). This research also intends to open the up the path for future studies, which can make alternative analyses about the topic or contemplate broader research, including other factors that have been little explored so far, i.e., referring to the adoption of lean management principles.

Regarding practical implications, by analysing the main determinant factors of eco-innovation, the aim is also to obtain useful information and knowledge for companies and to achieve added benefits not only for the company itself but also for all external partners involved in the innovation process. In addition, this analysis is intended to stimulate and guide companies in competitive processes in the scope of available incentive systems, aiming to increase investments in innovative activities related to sustainability, innovation, products, processes, organisational methods, and marketing, as well as the organisation and implementation of activities incorporating intensive knowledge and technology, to enable a better assessment the of application of the results of Research, Development and Innovation (RD&I) in terms of producing goods and providing services.

The set of empirical evidence obtained here contributes to the literature on the determinant factors of eco-innovation and sustainable business models, revealing the importance of forming cooperative relationships with competitors, universities, or other higher education institutions (for the total sample and the subsamples of high-tech and low-tech companies). New, innovative evidence is presented about different ways in which lean management principles influence the pro-eco-innovation orientation positively, regarding productivity (for the whole sample) and quality (e.g., high-tech subsample), and have a negative influence on the pro-eco-innovation orientation, regarding quality (e.g., low-tech subsamples).

The article is structured as follows. A review of the literature on eco-innovation and lean management principles is followed by exploring the determinant factors of eco-innovation, the identification of the research hypotheses, and the presentation of the proposed conceptual model. Then, the empirical study is presented with a discussion of the results. Finally, the conclusions are presented together with the limitations and suggestions for future research.

## 2. Literature Review

### 2.1. Eco-Innovation

Eco-innovation appeared for the first time in the literature about 20 years ago [20], but despite a growing interest in the topic, research in this field is still scarce due to the limited number of authors addressing the subject up to 1990 [21]. The term eco-innovation was first used by Fussler and James [22] in the book entitled "Driving Eco-Innovation: A Breakthrough Discipline for Innovation and Sustainability".

Later, in the mid-2000 s, eco-innovation became established as a popular subject in the scientific community [21], and subsequently, in 2008, it was recognised by the business community [23]. Although a relatively new concept [24], it has been approached from different angles of research, namely, innovation, management, engineering, and economics, among others [20].

For Porter and Linde [25] and Rennings [26], eco-innovations differ from other innovations in that the externalities and motivators for their introduction are involved, highlighting principally the importance of regulation policies for their development.

Therefore, eco-innovation can be defined as all processes, techniques, practices, systems, and products that are new or modified and which avoid or reduce environmental damage [27].

This concept also includes all changes in the product portfolio or production processes that follow sustainability goals, such as waste management, eco-efficiency, the reduction of emissions, recycling,

eco-design, or any other action implemented by companies that goes towards measuring and, then, reducing their ecological footprint.

Hellström [28] shares the views of Rennings [26], according to which eco-innovation is a process to develop new ideas, behaviour, products, and processes that contribute to reducing environmental effects or to ensure a greater harmony with environmental goals.

Andersen [29] adopts an evolutionary perspective of industrial dynamics and defines eco-innovation as an innovation able to attract green income in the market, focusing research efforts on the degree of integration of environmental questions in the economic process.

In this study, we adopt the definition of eco-innovation originally presented by Kemp and Pearson [30] and complemented by Horbach et al. [17] stating that eco-innovation is the production, application, or exploitation of goods, services, production processes, organizational structures, or management methods that have a novelty character for the company or user throughout its life cycle, representing the reduction of environmental risks and pollution, including a reduced negative impact of resource use, for example, energy, compared to relevant alternative options. To this definition, we couple the adoption of lean principles; advancing with our own definition thus exposed that eco-innovation is the production, application, or exploitation of goods, services, production processes, organizational structures, or management methods that are new to the company or user and exposed that, on an ongoing basis, we must focus on increasing efficiency, flexibility, and productivity, as well as eliminating more waste, following the logic of circular sustainability.

Reid and Miedzinski [30] consider eco-innovation as the creation of new, competitive efforts in products, processes, systems, services, and procedures conceived to satisfy human needs in order to improve the quality of life, through a minimal use of the lifecycle of natural resources, implying a minimal release of toxic substances.

Arundel and Kemp [31] underline that the benefits of eco-innovation should be accompanied by a change in company values in the same way that the reduction of environmental impacts requires a change in company management.

According to the OECD [32], eco-innovation is, first of all, an innovation in which the concept reflects explicitly the emphasis placed on reducing environmental impacts, whether intentionally or not, and, secondly, is not limited to innovations in products, processes, and organisational methods but also includes innovation in terms of social and institutional structures. Sarkar [33] also includes all forms of (technological and non-technological) innovation, new products, services, and new business models that contribute to developing new business opportunities that protect the environment.

Eco-innovation is also positioned as a wide-ranging way to address future environmental problems, through reducing energy, resources, waste, and consumption, tending to stimulate sustainable economic activities [2,28]. This view recognises that eco-innovation contributes to creating eco-companies [34] and can, therefore, be defined as a subclass of innovation, aiming to improve economic and environmental development [35,36].

According to Eurostat [37], the concept of eco-innovation has been subject to review, considering here all services, products, resources, and processes that go towards lessening some of the most significant environmental impacts, simultaneously optimizing natural resources. Eco-innovation reduces greenhouse gas emissions, can stimulate the use of recyclable materials, and favours the implementation of more environmentally friendly production processes and services.

For Nuij [38] eco-innovation is a response by the industry and the academic community to the development of new products and services in order to satisfy consumers in more eco-efficient contexts.

For Sarkar [33], the resulting benefits can be classified as direct and indirect. Considered as direct benefits are all the operational advantages, i.e., corresponding to economic gains arising from a more effective use of resources and better logistics. In turn, indirect benefits include an improved company image; relationships with suppliers, clients, and authorities; as well as strengthening the business innovation capacity in general.

### 2.2. Lean Management Principles

In the business context, the Lean paradigm is associated with companies' need to become increasingly competitive by reducing costs. Here, the focus is essentially on eliminating waste over the whole process, whether material flows or information flows, and, in this way, seeking to increase profitability, flexibility, and quality in processes. Consequently, the aim is to do more and better efficiently.

After the Second World War, in the Japanese car industry, more specifically in the Toyota Motor Corporation, the Lean thinking revolution appeared, aiming to optimize the production system. In its assembly lines, Toyota implemented the Toyota Production System (TPS) model, which was based on the philosophy of continuous improvement and focused on eliminating waste and reducing production times.

The TPS concept, designated Lean, was popularized in the book "The Machine that Changed the World" by Womack et al. [39], becoming one of the most quoted references as a paradigm of modern production in the last decades [40,41]. However, the Lean concept can still seem confusing and ambiguous, with managers, consultants, and academics specialized in the subject agreeing that a common, clear, consistent, and widely accepted definition does not exist [42].

According to Womack and Jones [43], the *Lean Thinking* paradigm, i.e., *Lean* thinking is centred on the continuous search for and elimination of all waste, aiming for an organisation's continuous improvement. Sayer and Williams [44], for example, defined it as being a holistic, sustainable approach aiming to maximize customer satisfaction. For Wilson [45], *Lean* is a set of techniques that, when combined and perfected, allow the reduction and elimination of waste, promoting organisations' flexibility and response capacity. Therefore, the *Lean* concept is applicable at all stages of a value chain, and the complete productive system, whether manufacturing a product or providing a service, can produce waste without affecting the value added for the target customer.

The reduction or elimination of waste alone is not an easy process, and for this reason, in Reference [44], it is argued that the lean paradigm should follow the following five principles: (i) value; (ii) value chain; (iii) flow optimization; (iv) implementation of a pull system; and (v) seeks perfection. Following these principles not only specifies the value of a product in precise terms and that this is really what the customer wants but also, through the value chain, identifies and analyses the flow of value for each product in the sense of being able to map activities that do not add value. Moreover, lean principles suggest that, after identifying the value chain and waste, a continuous flow must be created that is characterized by the ability to produce only what is needed for the moment. Also, regarding the principles to be followed by lean, only the customer's requests should trigger all the processes. Thus, organizations cannot produce what they think the customer will need but what is actually requested and in the exact quantity and moment, and the fact that, by encouraging a continuous improvement at all levels of the organization through the continuous listening of the client and the speed of responses, it will be possible to operate a continuous improvement of the organization [46]. The principles to be followed by the lean paradigm are fundamentally what will allow an operation in the elimination of waste, since this is the essential focus of this paradigm. After identifying and briefly characterizing the five basic principles of lean philosophy, a diagram (see Figure 1) is presented here to provide a better understanding of the link between the lean principles and the measurement measures adopted in this study in order to be able to assess the influence of a lean management principles adoption on the pro-eco-innovation orientation.

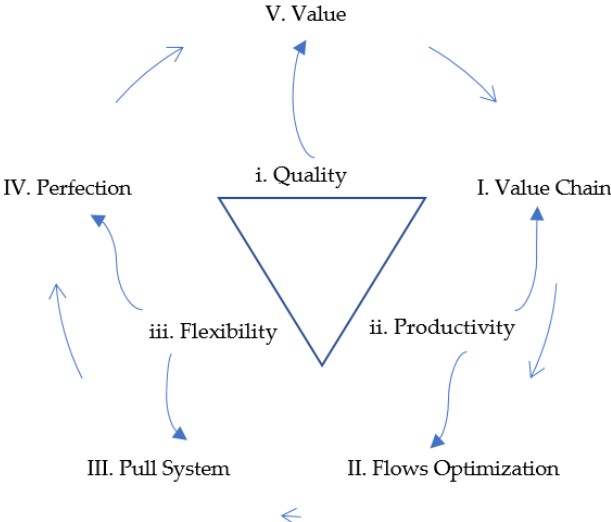

**Figure 1.** The lean principles and measures.

Given the intensification of competition coupled with the fact that markets have become increasingly global, companies have realized that, in order to become more competitive, it is no longer enough to improve efficiency within the organization itself and that it is, therefore, necessary to focus on the lean management of the entire value [47]. Therefore, as recommended in Figure 1, the adoption of the five principles of lean management is based on the creation and management of a value chain (I), which incorporates the flows optimization (II), in the sense of being implemented a true pull system (III), tending to perfection (IV) and thus guaranteeing the creation of processes that generate value added (V). The management of lean principles requires the operationalization of measures that, in the present research, are represented through (i) quality (to improve the quality of goods and/or services by guaranteeing the highest quality of the components of the chain of value (I) and the maximum value added (V)); (ii) productivity (to increase the production capacity of goods and/or services in order to ensure the maximization of the tradeoff between the results and resources allocated throughout the entire value chain (I) and flow optimization processes (II)); and (iii) flexibility (to improve flexibility in the production of goods and/or services to ensure a greater capacity of management and response of the organization in terms of the pull system (III) and the perfection of processes and productions (IV)).

The new approach proposed here is not only a formal mechanism for implementing a lean culture in organizations but also for strengthening the pro-eco-innovation orientation, which can be influenced by achieving positive outcomes such as increased quality, productivity, and flexibility. This perspective also allows us to argue that, despite quality, productivity, and flexibility being associated with all lean principles, the selected measures can be linked to different principles, adopting a lean philosophy of maximum value added based on a dual problem: the minimization of waste and the maximization of the triad formed by quality, productivity, and flexibility.

Thus, as shown in Figure 1, the adoption of the five lean principles in an organizational context requires the implementation of management practices that use lean measures represented by quality, productivity, and flexibility. Although the latter are an incomplete representation of these principles in view of the limited access to more detailed information, in the context of the present study, these three lean measures are used in a pioneering way to measure and test the influence of the adoption of lean principles on the pro eco-innovation orientation of companies with different levels of technological intensity.

### 2.3. Eco-Innovation Determinant Factors: Research Hypotheses and Conceptual Model

Many companies have implemented changes from a perspective oriented towards the environment, which in turn have a direct impact in different areas, namely R&D, production, administration, logistics, marketing, sales, and even globally in the value chain. Making the environmental question endogenous to firms' decision-making process stimulates the search for innovative activities that, to a certain extent, allow an appropriate coexistence of economic activities and the conservation of environmental resources and services.

From the vision presented above, for Van Den Bergh et al. [48], the main challenge is to ensure the management of production, distribution, and consumption, using renewable resources within their regeneration capacity and nonrenewable ones in accordance with the stage in their lifecycle and absorptive capacity in relation to the surrounding context. In this complex scenario, firms use eco-innovations to reduce the negative impacts of new processes or products on the environment, adopting sustainable marketing strategies and business models.

For a better understanding of the determinant factors of eco-innovation, some theoretical approaches identified in the literature follow, serving as cornerstones of this study. The small differences identified between authors arise mainly from how each one classifies the determinant factors, with being no substantial differences conceptually. Therefore, three types of determinant factors can be identified, namely (1) market-oriented factors such as market participation, competition, seeking new markets, labour costs, organisational image, and consumer demand; (2) technology-oriented factors, such as product quality, the efficiency of the material, product movement, and energy efficiency; and (3) factors caused by regulations, including current environmental legislation, occupational health and safety norms, and public and regulatory policies.

Although innovations are socially desirable, market imperfections can form barriers to development by private actors. When determinant factors stimulated by technology or by the market are not sufficiently strong, ecological innovations need to count on reinforced regulatory policies in order to ensure the desired dissemination [49].

Bernauer et al. [50] propose a conceptual framework to study the determinant factors of eco-innovation, dividing them in three groups:

- Regulatory: emphasizing questions related to the rigour of current environmental regulations and foreseeable changes in the future;
- Market: placing the emphasis on competitiveness and benefits provided to the consumer; and
- Internal to the company: highlighting "green" capacities, the capacity for business innovation and company size.

Horbach [51] carried out a study applied to the German context and proposed an alternative classification, distinguishing between

- Factors on the supply side: including technological capacities (based on human capital and knowledge) and problems in appropriating the results of innovations founded on restrictive market structures (for example, monopoly), company size, and scale gains;
- Factors on the demand side: considering the expectations of market demand, the development of environmental consciousness, and the preference for environmentally friendly, sustainable products; and
- Political and institutional factors: covering environmental policies oriented towards stimulating innovation based on incentives or approaches of institutional regulations and structure, i.e., on the political opportunities of environmentally oriented groups, the organisation of information flows, and the existence of innovation networks.

Using a database of companies located in the United Kingdom, Kesidou and Demirel [52] revealed that demand factors influenced investment decisions in eco-innovation, allowing the

alignment of commercial practices with social and economic expectations, consumers' requirements, and organisational capacities related to the existence of a system of environmental management, as well as corresponding to the rigour of environmental regulation policies. Regarding the last aspect, the same authors underline that the rigour of regulations affects differently the eco-innovations of less innovative companies compared to those of more innovative ones.

Horbach et al. [17] made important contributions to the issue of the determinant factors of eco-innovation in a quantitative analysis applied to Germany, which identified the factors determining eco-innovation according to the type of environmental impact, something which had not been explored in previous studies. These authors considered as determinant factors (i) the regulations (based on the previous study by Popp [53], where the national regulation is indicated as the main determinant factor of eco-innovations in the United States of America, Japan, and Germany); (ii) the factors stimulated by the market, referring to the contribution of Kammerer [54], which highlights the importance of the benefits for the consumer and the tacit recognition of the lack of strong incentives for eco-innovation on the demand side besides regulations in order to overcome the problem of double externality; (iii) the factors stimulated by technology (highlighting companies' technological capacities and environmental management systems); and (iv) the firm-specific factors (considering knowledge transfer mechanisms and an involvement in the relationship networks [55], as well as "green" capacities [54,56].

In the view of Horbach et al. [17], the still scarce literature on the determinant factors of eco-innovation has a certain complexity in relation to the mapping of supply, demand, or firm-specific factors. However, these authors emphasize the role of regulations, cost reduction, and the benefits for consumers. Current and expected regulations have effects on organisations concerning innovations related to gas reduction, water pollution, noise emission, and restrictions on dangerous substitutes, as well as on the increased possibility of recycling products. Cost reduction is important to motivate the reduction of energy and the use of materials, indicating the price of energy and materials, as well as taxation, as the main stimulants of eco-innovation. Consumers' requirements as a source of environmental innovations are related to the improved environmental performance of products and processes that increase the efficiency of materials and reduce energy consumption and waste, besides limiting the use of dangerous substances.

In the conceptual framework of the determinant factors of eco-innovation, the contributions of Horbach et al. [17] and Horbach [51], in the context of small and medium-sized enterprises (SME), indicate the predominance of incremental technology in the majority of environmental innovations implemented, based on limited research and development (R&D) efforts.

According to Kesidou and Demirel [52], organisations' behaviour in matters of a social, ethical, and juridical nature improve the company image but not necessarily environmental matters but can have a positive effect in large firms [55]. The certification of origin is not relevant in determining eco-innovation, since products originating from certified regions are protected by their reputation and the very concept, restricting preferences and the competition from substitute products considered unauthentic.

In relation to the role played by the allocation of public resources to promote eco-innovation, this study does not find a consensus in the literature of reference. For example, Horbach [51] and De Marchi [57] defend the positive influence of allocating this type of resource to promote eco-innovation. However, this perspective is refuted, among others, by Kammerer [54] and Triguero et al. [58]. This divergence of perspectives and results provided the opportunity for a revision of the theory to examine whether the triple helix model of Etzkowitz and Leydesdorff [59], which considers the integration of state, industrial, and academic efforts to promote innovation, is valid for eco-innovations in any circumstances or only under specific conditions.

In the study by Triguero et al. [58] about the determinant factors of different types of eco-innovation in the context of European SMEs, the following factors were considered:

- Supply side: the determinant factors are divided in those stimulated by technology (technological and management skills, cooperation with R&D institutes, agencies and universities, and access to

external knowledge and information) and by cost reduction (company size, the price of materials, and energy costs);

- Demand side: the factors are segmented according to those that are market-pulled (market participation and market demand for green products); and
- Factors pushed/pressurized through regulations: existing regulations, future regulations, and access to existing subsidies and tax incentives.

More recently, Cuerva et al. [60] carried out an application for Spanish food and drink companies, aiming to identify the differences between determinant factors of "green" and "non-green" innovations. The results reveal that the implementation of environmental management systems (EMS) and differentiation only explain the adoption of innovative green activities. These authors emphasize that technological capacities, such as R&D and human capital, promote smaller innovations than other innovations, contributing to the confirmation of previous empirical evidence [55,61–63] which highlights that financial restrictions limit green innovations to a greater extent than other innovations, as well as the development of suitable organisational capacities and the implementation of quality management systems aiming to promote ecological innovations to a greater extent than other innovations. Concerning R&D activities, the work by Trigueiro et al. [64] analysing synergies between eco-innovation and employment using a sample of more than 6000 innovative Spanish manufacturing and service firms should be noted. The results confirmed their positive influence on eco-innovation. Also in relation to R&D activities, namely internal R&D activities, the results presented in Reference [13] confirm that, when companies acquire knowledge from internal sources, this leads to increased innovation and sustainable performance.

Cuerva et al. [60] also underline that, in the SME context, practices of environmental responsibility and certification of origin do not have a positive influence on "green" innovations, although the contrary is true for conventional innovations.

Also according to Cuerva et al. [60], the cooperation among competitors, suppliers and clients, research centres, and universities does not have a significant influence on any type of innovation. Cooperation in the SME context discourages innovation in industries providing homogenous products. The results obtained by the authors open a window of opportunity to deepen the theory of open innovation of Chesbrough [65], who defends the formation of cooperative relationships between the actors of innovation in order to investigate in what circumstances and to what extent this applies to environmental innovations. Also based on open innovation, Lee et al. [66] empirically show the positive impact of external sources of information on innovation.

It should also be noted that, according to Cuerva et al. [60], company size has a positive influence on both types of innovation, confirming the empirical evidence of Cleff and Rennings [67], Bernauer et al. [50], De Marchi [57], and Le Bas and Poussing [68]. Therefore, the extra-financial resources necessary for innovation are also reason enough to suppose that large companies are more likely to innovate ecologically than small ones, above all because large firms are more likely to take on risks than SMEs. Nevertheless, some studies do not support that positive influence of company size on innovation [50,69], although recently, the work by Trigueiro et al. [64] has also contributed to the current debate, defending the influence of company size on eco-innovation.

The general theory of innovation usually emphasizes the relevance of the technological impulse as one of the main determinant factors of eco-innovation. The technological capacities available lead to eco-innovations. Also in the economic literature, technological and management capacities are generally considered to increase environmental innovations and the importance of technical knowledge acquired from external sources [51,58]. In this respect, Cohen and Levinthal [70] argue that the company's absorptive capacity or its ability to recognize the value of new, external information is extremely important in determining its innovation capacity. Therefore, the absorptive capacity will also provide companies with the resources necessary to recognize the potential of eco-innovation and to achieve their development [71].

From the above, the following research hypotheses are considered:

**H1.** *Technology has a positive influence on the pro-eco-innovation orientation.*

**H1a.** *Internal R&D activities have a positive influence on the pro-eco-innovation orientation.*

**H1b.** *External R&D activities have a positive influence on the pro-eco-innovation orientation.*

**H1c.** *An investment in computer equipment and software has a positive influence on the pro-eco-innovation orientation.*

**H1d.** *The acquisition of external knowledge has a positive influence on the pro-eco-innovation orientation.*

**H1e.** *Practices of Business Process Management (BPM) have a positive influence on the pro-eco-innovation orientation.*

**H1f.** *Work organisation practices have a positive influence on the pro-eco-innovation orientation.*

**H1g.** *Sources of information have a positive influence on the pro-eco-innovation orientation.*

The changes observed in market trends are generally related to eco-innovation opportunities. Suppliers and clients reinforce the need to develop this type of innovation [72]. Various studies observe that clients' perceptions or requirements can explain the firm's decision to adopt eco-innovations [17,73–75]. For example, Tsai et al. [76] analyse the growing tendency to purchase green, educational toys for children. Customers have a strong environmental conscience and are concerned about protecting the environment and producing a positive network externality effect, which can mean increased demand for green toys. In addition, manufacturers are willing to adopt the perceived value of green toys for customers if they can handle the difficulties of a cooperation within the production chain and production. A usual policy recommendation is to reduce the financial restrictions for SMEs, with the final aim of encouraging eco-innovation [60]. Johnson and Lybecker [77] explore the forms of public and private financing, attempting to assess its effectiveness and making policy suggestions oriented to financing eco-innovation. In private companies, there are growing difficulties in the emergence of eco-innovations compared to other innovations to attract risk capital for their development [78]. Therefore, the availability of finance is considered one of the main stimulants of eco-innovations [77]. Pressure groups or stakeholders have also been highlighted as alternative forces that influence business in eco-innovation practices.

Guoyou et al. [79] observe that foreign clients have a critical role in leading companies to adopt eco-innovation strategies in processes and products, although foreign investors only affect the adoption of eco-innovators.

This results in the following research hypotheses:

**H2.** *Market characteristics have a positive influence on the pro-eco-innovation orientation.*

**H2a.** *Cost savings have a positive influence on the pro-eco-innovation orientation.*

**H2b.** *New markets have a positive influence on the pro-eco-innovation orientation.*

**H2c.** *Market participation has a positive influence on the pro-eco-innovation.*

Jänicke [80] argues that intelligent regulations plays an important role in the political competition for eco-innovation and can be identified as a driving force of eco-innovation. Therefore, the argument that regulation imposes high costs on firms and hinders innovation and stronger competitiveness has remained popular. At the beginning of the 1990s, however, the defendants of regulations successfully challenged the neoclassical tradition, emphasizing the existence of a positive relationship between environmental regulations and countries' competitiveness. The same author argues that environmental regulations can create barriers for companies and industries, but they can also provide a number of advantages.

Despite the important contribution of regulation to creating and spreading eco-innovation, as shown in the literature, Mickwitz et al. [81] argue that regulations, i.e., environmental norms and licensing conditions, have often been considered ineffective in inducing innovations. The basic, underlying argument is that regulations do not give any additional incentive to innovate after fulfilling

requirements. Based on theoretical models, many economists argue that economic instruments are more useful that regulations in promoting innovation because they impose a cost for pollution, irrespective of its level, and are therefore an incentive to innovate [82,83]. This occurs because reduced emissions provide cost savings in the form of taxes avoided or increase income in the form of subsidies obtained or licences that can be sold. Therefore, the following research hypotheses are considered:

**H3.** *Policies have a positive influence on the pro-eco-innovation orientation.*

**H3a.** *European, public lines of finance have a positive influence on the pro-eco-innovation orientation.*

Rusko [84] argued that one of the main motivations for competitors to become involved in strategic cooperation agreements is based on benefiting the creation of added value and, thereby, improving economic performance. Therefore, gaining a greater added value and creating a bigger market are important factors for competing partners to become involved in this type of relationship [84–86].

As proposed by Padula and Dagnino [87], the cooperation strategy is influenced by the company's knowledge structure. Firms wishing to become involved in such cooperation relationships face the need to acquire or increase their stock of internal knowledge resources [88,89]. Acquiring knowledge externally is extremely important for companies in order to strengthen competitiveness and innovation when they compete with their rivals. This results in the following research hypotheses:

**H4.** *Cooperation relationships have a positive influence on the pro-eco-innovation orientation.*

**H4a.** *Cooperation with competitors has a positive influence on the pro-eco-innovation orientation.*

**H4b.** *Cooperation with universities has a positive influence on the pro-eco-innovation orientation.*

The lean philosophy and eco-innovation are often seen as compatible due to their combined focus on waste reduction. The removal of non-value-added activities suggested by the lean paradigm can provide substantial energy savings and can reduce the environmental impact of production systems by identifying opportunities for the integration of lean efforts and innovations [90].

For Elias and Magalhães [91], the tools developed by *lean* management contribute to gaining environmental, technological, and economic benefits, thereby minimizing the need for resources, energy, y and raw material.

According to Porter and Linde [25], when applying *lean* management principles, an organisation influences the proactiveness with the purpose of adopting eco-innovation in firms, or also, the more generalized *lean* production is in those organisations, the less the waste and costs are associated with the production process. Therefore, implementing a no-waste production policy in organisations influences, to some extent, the development of innovation activities, promoting a change in organisations' performance in that area.

At the same time, since *lean* initiatives can ensure the necessary production flows, only a small amount of stock should be obtained, produced, transported, packaged, and handled, which also minimizes the negative environmental impacts. Regarding quality, ISO 14001 certification is the basis of a systematic approach to reduce organisations' environmental impacts and, at the same time, acts as a snowball that influences organisations to adopt eco-innovative practices.

Considering that companies continually strive to optimize their operations in general, including the product development process, [92] it is argued that a robust design can be considered as an integral part of such a product development. Following Reference [92], the robust design has as a preferential focus the action of designing products with a functional performance insensitive to variation and noise. Thus, the importance of a robust design in lean management principles is recognized; however, it still seems challenging to operationalize, in empirical terms, what will be its influence in the pro-eco-innovation orientation.

It should also be underlined that there are trade-offs to consider when the *lean* paradigm is associated with the determinant factors of eco-innovation. The truth is that *lean* strategies involving *just-in-time* deliveries of small amounts can mean greater transport, packaging, and handling costs,

which may contradict an eco-innovative approach. Recognising this conflict, companies can identify compromises or develop solutions that lessen undesirable consequences. For example, firms that recognise the negative environmental impact of the *just-in-time* approach can consider reusable containers and packaging or can adapt the size of the batch to optimize transport use as a means to achieve *lean* objectives and eco-innovation. In this line of reasoning, in some cases, *lean* can be considered a determinant factor of eco-innovation, as will be seen further on from the results obtained in the empirical analysis. Therefore, the following research hypotheses are presented:

**H5.** *Lean management principles have a positive influence on the pro-eco-innovation orientation.*

**H5a.** *Quality management has a positive influence on the pro-eco-innovation orientation.*

**H5b.** *Flexibility has a positive influence on the pro-eco-innovation orientation.*

**H5c.** *Productivity has a positive influence on the pro-eco-innovation orientation.*

Although the current approach to eco-innovation was founded on a previously tested set of determinant factors [17], grouped by the dimensions technology (H1), market (H2), public policies (H3), and cooperative relations (H4), it is worth noting that this approach differs from the previous ones, since the first technology dimension represented in H1 is based on metrics of internal R&D activities, external R&D activities, equipment, software, external knowledge, business process management, labour organization, and sources of information. In relation to the market represented in H2, distinct metrics are used: cost savings, entry into new markets, and market share. When addressing public policies represented in H3, these are measured by the provision of funding lines, and with regard to the cooperation represented in H4, it is considered, above all, the relations established with universities, research structures, and competitors. The novelty here lies in the formulation and test of the dimension still unexplored of the lean principle adoption represented in H5, making use of success critical variables, such as quality, productivity, and flexibility, used to reinforce the pro-eco-innovation orientation.

Considering the review of the literature of reference and setting out from the proposal of Horbach et al. [17], a conceptual model is proposed (Figure 1), which is of an innovative nature inasmuch as it includes four groups of determinant factors present in the literature, namely technology, market, public policies, and cooperation relationships, added to which is a fifth group of determinants still to be explored: lean management (see Figure 2 below).

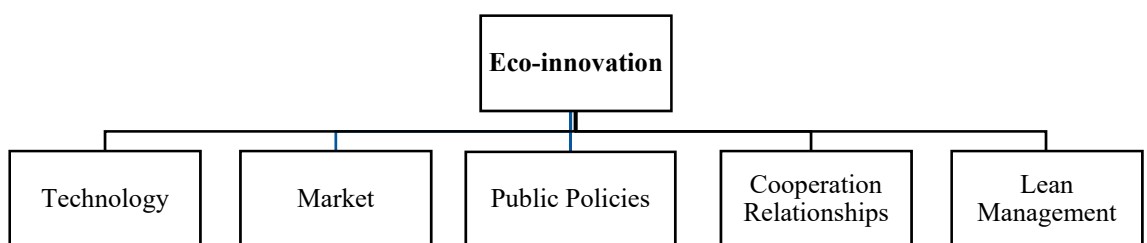

**Figure 2.** The determinant factors of eco-innovation. Source: Own elaboration.

## 3. Empirical Study

### 3.1. Database and Sample

This study is based on a dataset obtained from the Community Innovation Survey—CIS 2010 (Community Innovation Survey 2010). The sample is created by the Portuguese National Statistics Institute (INE), with the process being carried out by the Office of Planning, Strategy, Assessment, and International Relations of the Ministry of Science, Technology, and Higher Education (GPEARI/MCTES), under EUROSTAT supervision. The methodology used in this research agrees with that described in the OECD's Oslo Manual and is adopted everywhere in Europe through EUROSTAT [93].

The CIS 2010 questionnaire provides detailed information about firms' general data, namely the sector of activity, the number of employees, staff training and qualifications, investment and expenditure regarding R&D activities, turnover, cooperation, and public financial support.

The sample obtained, after being corrected from the survey results, contained 8.189 Portuguese companies, being designated the corrected sample. Of this sample, 6160 companies answered the questionnaire, corresponding to a response rate of 76% [94]. This sample is formed of firms with at least 10 employees, and when the firm has 250 or more employees, it is subject to an exhaustive survey. The sample was built by the INE according to the methodological specifications of EUROSTAT. It was stratified according to a 2-digit CAE, the size (considering the number of employees), and the regional distribution (NUTS II). However, in the current analysis, due to the lack of data and sector classification carried out based on EUROSTAT for NACE Rev.2—a 2-digit classification—only a sample of 334 companies have all the data for the selected variables. Therefore, the 334 companies correspond to the total number of valid cases for which the complete information about the set of independent variables studied is available. Besides the sector classification, the total sample was divided into subsamples for "high-tech" and "low-tech", with 95 and 234 companies respectively (Table 1).

**Table 1.** The distribution of companies by sector classification: EUROSTAT NACE Rev.2.

| Classification | Description | NACE Rev.2 | No. of Companies |
|---|---|---|---|
| High-Tech | Manufacture of chemical products. | 20 | 95 |
| | Manufacture of basic pharmaceutical products and pharmaceutical preparations. | 21 | |
| | Manufacture of computers, electronic products, and optics. | 26 | |
| | Manufacture of electrical equipment; machinery and equipment n.e.c; motor vehicles, trailers, and semitrailers; and other transport equipment. | 27; 28; 29; 30 | |
| Low-Tech | Manufacture of food products, beverages, tobacco products, textiles, clothing, leather and related products, wood and wood products, paper and paper products, printing, and reproduction of media. | 10; 11; 12; 13; 14; 15; 16; 17; 18 | 239 |
| | Manufacture of coke and refined petroleum products. | 19 | |
| | Manufacture of rubber and plastic products, other nonmetallic mineral products, and metal products manufactured except machinery and equipment. | 22; 23; 24; 25 | |
| | Manufacture of furniture; Other manufacturing industries. | 31; 32 | |
| | Repair and installation of machinery and equipment. | 33 | |
| Total | | | 334 |

To better understand our total sample as well as the subsamples of high-tech and low-tech companies, we present here its composition with respect to the company size (Table 2). Therefore, the total sample has 334 companies, of which 253 companies have below 50 workers and 81 companies have 250 or more workers, and in turn, the subsample of high-tech counts a total of 95 companies, among which 26 have below 50 workers and 69 companies have 250 or more workers. The subsample of low-tech counts on 239 companies, among which 184 has less than 50 workers and 55 companies have 250 and more workers.

**Table 2.** The distribution of company size by the total sample and subsamples.

| | Total | High-Tech | Low-Tech |
|---|---|---|---|
| Number of companies | 334 | 95 | 239 |
| Company size | | | |
| Under 50 workers | 253 | 26 | 184 |
| 250 workers or more | 81 | 69 | 55 |

The sample and the subsamples underwent a logistic regression process for the purpose of estimating the conceptual model proposed and an empirical testing of the research hypotheses and sub-hypotheses for the different determinant factors of eco-innovation. As already mentioned, the dependent variables correspond to the eco-innovation of products or processes (equal to 1 for companies that carried out product or process innovation, and 0 otherwise), which concerns companies that created and introduced new products or significantly improved processes. In addition, all the independent and control variables are binary.

### 3.2. Variables and Model Specification

This study focuses on analysing the determinant factors of eco-innovation. Therefore, the variables "Reduce the environmental impact" (OREI) and "Reduce the material and energy used per unit produced" (ORME), with the original designation (in brackets) of the CIS 2010 variables, are the dependent variables. These are binary variables analysing whether the company, in the period 2008–2010, introduced innovative products or processes, i.e., equal to 1, if it introduced new or significantly improved products or processes, and equal to 0 if it did not introduce any type of product or process innovation with the objectives defined according to OREI and ORME. Both dependent variables, which are possible to select in the CIS 2010 database, ORME and OREI, make it easier to understand what we consider as a pro-eco-innovation orientation, which is precisely the organizational orientation for the adoption of new products and new processes that aim at improving efficiency, flexibility, and productivity, following a circular sustainability logic.

As for the independent variables, this research uses the variables associated with five factors: (1) technology, (2) market, (3) public policies, (4) cooperation relationships, and (5) lean management principles. Besides the dependent and independent variables, the control variables of size (SIZE_3), group headquarters (c_Ho), and group (GP) are included in this research.

For the size variable (SIZE_3), the analysis is made considering the European classification, i.e., companies with up to 10 employees are considered micro-companies, while small companies have between 10 and 49 employees and medium-sized ones have between 50 and 250 employees. With over 250 employees, companies are considered large. For the group headquarters variable (c_HO), the possible influence of a group headquarters location is analysed and the group variable (GP) lets us determine the influence of belonging to a group of companies. In this connection, LeBas and Poussing [68] argued that belonging to an economic group has a positive influence on the probability of innovation, proposing that this effect can be related to a greater availability of resources and the stability of R&D expenditures in firms belonging to a given economic group, considering the specificity and complexity of environmental technology [95]. As already mentioned, in order to estimate the proposed model and empirically test the research hypotheses and sub-hypotheses for the different determinants of eco-innovation, the present study adopts the logistic regression model due to the need to analyze the statistical relationship of a binary dependent variable in relation to more than one explanatory variable, that is, how the independent variables influence companies in the creation and introduction of new products or processes significantly improved (e.g., eco-innovation of products or processes). The logistic regression model is present in empirical studies dealing with the same relationship as this research [96–101], and so, it is presented as a viable model to carry out this study. For Marôco [102], a logistic regression is an econometric method that is used to model the occurrence in probabilistic terms of one of the two achievements of the dependent variable classes, where the independent variables can be qualitative or quantitative. This method also allows the evaluation of the significance of each of the independent variables included in the model.

In other words, a logistic regression is a statistical technique that aims to produce, from a set of observations, a model that allows the prediction of values taken by a categorical variable, often binary, from a series of continuous explanatory variables and/or binary. In the regression analysis logistic model, the regression model of the probability of observing a given event is expressed as follows:

$$P_{estimate=\frac{e^L}{1+e^L}} \tag{1}$$

In equivalent terms:

$$P_{estimate=\frac{e^L}{1+e^{-L}}} \tag{2}$$

where

$$L = \beta_0 + \beta_1 * X_1 + \beta_2 * X_2 + \ldots + \beta_k * X_k \tag{3}$$

where $P_{estimate}$ is the estimated probability of a given event occurring; $\approx$2718, the Nepper number, is the value used in the exponential function; and $\beta_1, \beta_2, \ldots, \beta_k$ are the estimated regression coefficients that correspond to the $k$ independent variables, with $\beta_0$ being the estimated model constant. The parameters (coefficients) considered are estimated by means of the maximum likelihood method, which consists of determining the values of the parameters that maximize the probability of obtaining the set of observed values. According to Hosmer and Lemeshow [103], the maximum likelihood method allows for the estimation of the regression coefficients, which maximizes the probability of obtaining the realizations of the dependent variable of the sample. The likelihood function expresses the probability of observed data, such as unknown parameters. The maximum likelihood estimators of these parameters are chosen in order to obtain the maximum likelihood, being expressed in the following system of equations:

$$\begin{cases} \frac{\delta L}{\delta \beta o} = 0 \\ \frac{\delta L}{\delta \beta j} = 0 \end{cases} \Leftrightarrow \begin{cases} \sum_{i=1}^n [yi - \pi(xi)] = 0 \\ \sum_{i=1}^n xij[yi - \pi(xi)] = 0 \end{cases}, j = 1, \ldots, m \tag{4}$$

The logistic regression model is the most usual method [104] that makes use of the maximum likelihood estimation to evaluate the probability of categorical association [105], and the Results Section may be verified through the value obtained for the logarithmic likelihood as well as through the $p$-value obtained for assessing if the model accurately represents the data.

The variables presented in the conceptual model proposed are presented in Table 3.

Considering the information on the method and variables underlying this research, the logistic regression model was built, represented according to the two following specifications selected:

$$\begin{aligned}
ORME_i = \beta_0 &+ \beta_1 RRDIN + \beta_2 RRDEX + \beta_3 RMAC + \beta_4 ROEK + \beta_5 RTR \\
&+ \beta_6 INPSPD + \beta_7 INPSLG + \beta_8 INPSSU + \beta_9 ORGBUP + \beta_{10} ORGWKP \\
&+ \beta_{11} ORGEXP + \beta_{12} SSUP + \beta_{13} SCLI + \beta_{14} SCOM + \beta_{15} SINS \\
&+ \beta_{16} SUNI + \beta_{17} SGMT + \beta_{18} SCON + \beta_{19} SJOU + \beta_{20} SPRO \\
&+ \beta_{21} OLBR + \beta_{22} NEWMKT + \beta_{23} ONMOMS + \beta_{24} FUNGMT \\
&+ \beta_{25} FUNEU + \beta_{26} OQUA + \beta_{27} OFLEX + \beta_{28} OCAP + \beta_{29} CO \\
&+ \beta_{30} CO41 + \beta_{31} C042 + \beta_{32} CO61 + \varepsilon_i
\end{aligned} \tag{5}$$

$$\begin{aligned}
OREI_i = \beta_0 &+ \beta_1 RRDIN + \beta_2 RRDEX + \beta_3 RMAC + \beta_4 ROEK + \beta_5 RTR + \beta_6 INPSPD \\
&+ \beta_7 INPSLG + \beta_8 INPSSU + \beta_9 ORGBUP + \beta_{10} ORGWKP + \beta_{11} ORGEXP \\
&+ \beta_{12} SSUP + \beta_{13} SCLI + \beta_{14} SCOM + \beta_{15} SINS + \beta_{16} SUNI + \beta_{17} SGMT \\
&+ \beta_{18} SCON + \beta_{19} SJOU + \beta_{20} SPRO + \beta_{21} OLBR + \beta_{22} NEWMKT \\
&+ \beta_{23} ONMOMS + \beta_{24} FUNGMT + \beta_{25} FUNEU + \beta_{26} OQUA + \beta_{27} OFLEX \\
&+ \beta_{28} OCAP + \beta_{29} CO + \beta_{30} CO41 + \beta_{31} C042 + \beta_{32} CO61 + \varepsilon_i
\end{aligned} \tag{6}$$

where $ORME_i$ is the reduction in the material and energy used per unit produced; $OREI_i$ is the reduction in the environmental impact; $\beta$ are the coefficients; and $\varepsilon_i$ is the residual.

**Table 3.** The conceptual model variables.

| | Variables | | Description |
|---|---|---|---|
| *Dependent variables* | Process innovation | ORME | Reduce the material and energy used per unit produced |
| | Product innovation | OREI | Reduce the environmental impact |
| *Independent variables* | Technology | RRDIN | R&D activities carried out within the company |
| | | RRDEX | External acquisition of R&D activities |
| | | RMAC | Acquisition of machinery, equipment, and software |
| | | ROEK | Acquisition of other external knowledge |
| | | RTR | Training for innovation activities |
| | | INPSPD | New or significantly improved fabrication or production methods |
| | | INPSLG | Logistic, delivery, or distribution methods of new or significantly improved production factors or final products |
| | | INPSSU | New or significantly improved activities supporting the firm's processes |
| | | ORGBUP | Introduction of new business practices in organising procedures |
| | | ORGWKP | The firm introduced new methods of organising responsibilities and decision-making |
| | | ORGEXR | Introduction of new methods of organising external relations with other firms or public institutions |
| | | SSUP | Suppliers of equipment, material, components, or software |
| | | SCLI | Clients or consumers |
| | | SCOM | Competitors or other firms in the same sector of activity |
| | | SINS | Consultants, laboratories, or private R&D institutions |
| | | SUNI | Universities or other higher education institutions |
| | | SGMT | State laboratories or other public bodies with R&D activities |
| | | SCON | Conferences, fairs, and exhibitions |
| | | SJOU | Scientific journals and technical/professional/commercial publications |
| | | SPRO | Professional or business associations |
| | Market | OLBR | Reduce labour costs per unit produced |
| | | NEWMKT | Open new markets |
| | | ONMOMS | Increase market participation |
| | Public Policies | FUNGMT | Public financial support from central administration (including agencies or ministries, through government programmes) |
| | | FUNEU | Public financial support from the European Union |
| | Cooperation relations | CO41 | Competitors or other firms in the same sector of activity: Portugal |
| | | CO42 | Competitors or other firms in the same sector of activity: Other European countries |
| | | CO61 | Universities or other higher education institutions: Portugal |
| | Lean Management Principles | OQUA | Improve the quality of goods and/or services |
| | | OCAP | Increase the production capacity of goods and/or services |
| | | OFLEX | Improve flexibility in producing goods and/or services |
| | Control | SIZE_3 | Total number of people working in the firm in 2010 |
| | | C_GO | Country in which the group headquarters is located |
| | | GP | In 2010, the firm belonged to a group of firms |

## 4. Results

The logistic regression models were estimated for the total sample and for the subsamples of high and low technology, as identified in Tables 4–6. For each sample, the dependent variables identified (ORME and OREI) were used, representing the main motivations for a pro-eco-innovation orientation, i.e., increased productivity and reduced environmental impact.

Considering the total sample of 334 companies (Table 4), for the (ORME) dependent variable, the model represents the data with a statistical accuracy, in that the logarithmic likelihood of the model of −128.719 and a *p*-value of 0.000 were obtained, and therefore, the model is statistically significant. For the (OREI) dependent variable, the model is also statistically significant, with a logarithmic likelihood of the model of −151.841 and a *p*-value of approximately 0.000002.

**Table 4.** The logit model for the total sample.

| | Variables | Process Innovation | Product Innovation |
| --- | --- | --- | --- |
| | | ORME | OREI |
| Technology | RRDIN | −0.035 | −0.922 * |
| | RRDEX | −0.365 | −0.144 |
| | RMAC | 0.371 | 0.044 |
| | ROEK | −0.201 | −0.525 |
| | RTR | −0.776 | −0.314 |
| | INPSPD | −0.159 | −0.716 * |
| | INPSLG | 0.045 | 0.695 ** |
| | INPSSU | 0.323 | 0.151 |
| | ORGBUP | 0.563 | 0.183 |
| | ORGWKP | 0.740 * | 0.593 |
| | ORGEXR | 0.425 | 0.448 |
| | SSUP | 0.175 | 0.123 |
| | SCLI | 1.000 ** | 1.360 *** |
| | SCOM | 0.529 | −0.295 |
| | SINS | 0.211 | 0.837 ** |
| | SUNI | −0.467 | −0.383 |
| | SGMT | −0.045 | 0.321 |
| | SCON | 0.396 | 0.641 * |
| | SJOU | −0.319 | 0.650 * |
| | SPRO | 0.602 | 0.090 |
| Market | OLBR | 2.575 *** | 1.504 *** |
| | NEWMKT | −0.567 | −0.167 |
| | ONMOMS | −0.678 | −0.498 |
| Public Policies | FUNGMT | 0.514 | 0.206 |
| | FUNEU | 0.232 | −0.182 |
| Lean Management Principles | OQUA | −0.331 | −1.515 * |
| | OFLEX | −0.270 | −0.021 |
| | OCAP | 0.999 ** | 0.536 |
| Cooperation Relationships | CO41 | 0.758 | 0.106 |
| | CO42 | 1,079 * | −0.217 |
| | CO61 | −0.244 | −0.036 |
| Control | SIZE_3 | 1.044 ** | 1.587 *** |
| | C_HO | −0.835 | −0.117 |
| | GP | 0.250 | −0.029 |
| Constant | C | −2.847 | −0.523 |
| | Observations | 334 | 334 |
| | Log Likelihood | −128.718 | −151.840 |
| | Adjusted R$^2$ | 0.316 | 0.219 |

* a significance of 10%, ** a significance of 5%, and *** a significance of 1%.

Considering the subsample of 95 high-tech companies (Table 5), for the (ORME) dependent variable, the logarithmic likelihood is −24.94 and the *p*-value is 0.111, being similar for the (OREI) dependent variable with values of −23.870 and 0.030, respectively.

It is also concluded that the global model is statistically more accurate than the high-tech model, since the likelihood is greater. McFadden's likelihood ratio also reveals that figures from 0.2 to 0.4 are a good indicator of the model's quality being appropriate. As the models presented previously are within this interval, it can be considered a strong model.

In this case, the aim is to obtain a model representing the least error for the different variables, and for this, it was necessary to adjust the number of variables, removing the SIZE_3, C_HO, and GP variables, since McFadden's ratio was lower. Therefore, we now have a model of 32 instead of 35 variables. This is due to the sample being smaller and influencing the model's estimators.

**Table 5.** The logit model for the high-tech subsample.

|  | **Variables** | **Process Innovation** | **Product Innovation** |
|---|---|---|---|
|  |  | **ORME** | **OREI** |
| Technology | RRDIN | 1.023 | 2.769 |
|  | RRDEX | −0.278 | −4.706 * |
|  | RMAC | 2.697 | 4.692 |
|  | ROEK | −1.652 | −1.850 |
|  | RTR | −1.024 | 0.160 |
|  | INPSPD | −5.200 | 1.130 |
|  | INPSLG | 1.906 | 0.868 |
|  | INPSSU | −1.477 | −1.134 |
|  | ORGBUP | 1.714 | 1.482 |
|  | ORGWKP | 2.907 * | 0.554 |
|  | ORGEXR | 1.454 | 1.740 |
|  | SSUP | 1.816 | −0.484 |
|  | SCLI | −0.952 | 3.874 |
|  | SCOM | 1.452 | −0.745 |
|  | SINS | −0.255 | 0.116 |
|  | SUNI | −0.272 | −4.156 * |
|  | SGMT | 0.538 | 0.426 |
|  | SCON | 0.264 | 1.961 |
|  | SJOU | −3.593 | 2.460 |
|  | SPRO | 0.784 | −1.306 |
| Market | OLBR | −1.816 | −1.236 |
|  | NEWMKT | −0.333 | −0.128 |
|  | ONMOMS | 6.853 * | −0.375 |
| Public Policies | FUNGMT | 0.725 | 3.454 * |
|  | FUNEU | −0.518 | −2.250 |
| Lean Management Practices | OQUA | 7.191 * | −8.806 ** |
|  | OFLEX | −0.198 | −0.291 |
|  | OCAP | 1.691 | 3.271 |
| Cooperation Relationships | CO41 | −0.341 | 1.354 |
|  | CO42 | −0.902 | −2.272 |
|  | CO61 | 0.946 | 4.401 * |
| Constant | C | −10.358 | −2.329 |
|  | Observations | 95 | 95 |
|  | Log Likelihood | −24.194 | −23.87015 |
|  | Adjusted $R^2$ | 0.457 | 0.497 |

* a significance of 10%, ** a significance of 5%, and *** a significance of 1%.

For the subsample of 239 low-tech companies (Table 6), regarding the (ORME) dependent variable, the likelihood is −78.2138 and the *p*-value is 0. For the (OREI) dependent variable, the values of −108.481 and 0.000 are obtained respectively. The model for both dependent variables is, therefore, statistically significant.

**Table 6.** The logit model for the low-tech subsample.

| | **Variables** | **Process Innovation** | **Product Innovation** |
|---|---|---|---|
| | | **ORME** | **OREI** |
| Technology | RRDIN | 0.018 | −1.683 ** |
| | RRDEX | −0.289 | 0.298 |
| | RMAC | 0.477 | −0.300 |
| | ROEK | −0.372 | −0.987 ** |
| | RTR | −1.200 * | −0.260 |
| | INPSPD | 0.149 | −1.238 ** |
| | INPSLG | −0.148 | 0.898 ** |
| | INPSSU | 0.928 * | 0.269 |
| | ORGBUP | 0.688 | 0.017 |
| | ORGWKP | 0.707 | 0.771 |
| | ORGEXR | 0.421 | 0.346 |
| | SSUP | −0.112 | −0.104 |
| | SCLI | 2.195 *** | 1.831 *** |
| | SCOM | 0.951 * | −0.537 |
| | SINS | 0.198 | 1.394 *** |
| | SUNI | −0.564 | 0.476 |
| | SGMT | −0.371 | −0.044 |
| | SCON | 0.310 | 0.835* |
| | SJOU | 0.017 | 0.739 |
| | SPRO | 1.398 ** | 0.183 |
| Market | OLBR | 4.064 *** | 2.033 *** |
| | NEWMKT | −0.387 | 0.102 |
| | ONMOMS | −1,066 | −0.625 |
| Public Policies | FUNGMT | 0.625 | −0.287 |
| | FUNEU | −0.013 | −0.224 |
| Lean Management Practices | OQUA | −2.151 * | −1.070 |
| | OFLEX | −0.370 | −0.091 |
| | OCAP | 0.971 | 0.586 |
| Cooperation Relationships | CO41 | 1.054 | −0.388 |
| | CO42 | 2.251 ** | −0.181 |
| | CO61 | −0.619 | −0.807 |
| Control | SIZE_3 | 1.469 ** | 1.911 *** |
| | C_HO | −2.133 *** | −0.397 |
| | GP | 0.990 | −0.405 |
| Constant | C | −3.807 ** | −0.304 |
| | Observations | 239 | 239 |
| | Log Likelihood | −78.21376 | −102.480 |
| | Adjusted $R^2$ | 0.448 | 0.295 |

* a significance of 10%, ** a significance of 5%, and *** a significance of 1%.

## 5. Discussion

The results will now be discussed using the previously formulated research hypotheses to examine the role played by *lean* management principles in the scope of the determinant factors of eco-innovation for companies with different levels of technological intensity.

Considering hypothesis H1, predicting a positive effect of technology on the pro-eco-innovation orientation, this is confirmed for the total sample of companies and for the subsamples of high- and low-tech companies. H1 is not rejected, as there are significant variables standing out among them: the introduction of new methods of organising responsibilities and decision-making (ORWKP); activities supporting firms' new or significantly improved processes, such as new systems of maintenance, accounting, or computing (INPSSU); logistic, delivery, or distribution methods of new or significantly improved production factors or final products (INPSLG); information sources such as clients or consumers (SCLI); competitors or other companies in the same sector of activity (SCOM); and professional or business associations (SPRO).

This result is in line with previous studies, such as Horbach et al. [17], Horbach [51], Triguero et al. [58], Cuerva et al. [60], and Segarra-Oña et al. [106], which argued that technological factors orient eco-innovation. Knowing that information sources are associated with the technology factor, as in Triguero et al. [58], and observing the variables used in this study, they are found to have a significant and positive influence on technology, as demonstrated in the study by Lee et al. [66], which, based on open innovation, reveals empirically the positive impact of external sources of information on innovation. Pacheco et al. [107] provided a systematic literature review on eco-innovation determinants in manufacturing SMEs, identifying the available resources (e.g., people, technology, and know-how) as the most important determinant factor. This is justified by the fact that eco-innovation requires an investment in qualified human capital, as well as the acquisition of technology or knowledge.

At the same time, according not only to Trigueiro et al. [64] but also to Horbach [51], who indicates that technology, through R&D activities, triggers eco-innovations, the results obtained here confirm that R&D activities are significant but have a negative influence. Conversely, in the context of internal R&D activities, the previous empirical results obtained in Reference [13], also suggest that the internal acquisition of R&D activities leads to an increased innovation as well as to a sustainable performance.

Also in this context, Cuerva et al. [60] emphasized that, in a low-tech sector, technological capacities through R&D activities only promote conventional innovation. In addition, Triguero et al. [65] showed that belonging to a high-tech sector increases the likelihood of an occurring eco-innovation; this remark is ratified through the results now presented.

H2 predicts a positive influence of market characteristics on pro-eco-innovation. Here, a positive effect is confirmed in the total firm sample, and so, H2 is not rejected. The same results are found for both high-tech and low-tech companies. The most important variables in this group are the reduction of labour costs per unit produced (OLBR) and an entry to new markets or an increased market quota (ONMOMS). Regarding cost reduction, Scarpellini et al. [108] emphasized that eco-innovation projects, which aim to reduce costs and, therefore driven by an efficiency logic, tend to be oriented towards process innovation. The latter are more frequent in the context of companies that are not so labor intensive.

The results are in line with previous studies by Horbach et al. [17], Bernauer et al. [50], Horbach [51], Kammerer [54], and Ambec and Lanoie [109], which highlight that eco-innovation is also stimulated by the market factor.

Horbach et al. [17], Horbach [51], Kesidou and Demirel [52], and Triguero et al. [58] are examples of studies defending the proposition in the third research hypothesis (H3), i.e., public policies have a positive influence on eco-innovation. In this study, H3 is not rejected for the subsample of high-tech companies. However, for low-tech companies, there is no evidence of policy formulation having a significant influence on process and product innovation and, so, for low-tech companies, H3 is rejected. Still, regarding public policies, Scarpellini et al. [110] pointed out a positive relationship between public incentives and eco-innovation, arguing that the former would reduce the risk associated with investing in eco-innovation projects, contributing to an increased profitability. In the same line of thought, the new empirical evidence now presented supports the basic idea that public policies have a positive influence on eco-innovation.

For hypothesis H4, concerning cooperation relationships, a positive and significant influence on pro-eco-innovation orientation is discussed. In the analysis of firms in the total sample, high-tech firms, and low-tech firms, a positive influence of cooperation relationships was detected, namely with competitors and universities in Portugal, and so, H4 is not rejected. The variables underlying this non-rejection decision are the following: competitors or other companies in the same sector of activity, Portugal (CO41), and universities or other higher education institutions, Portugal (CO61). According to Hájek et al. [13], universities should establish cooperative relationships with companies in compatible research areas, in order to foster the practical application of both knowledge transfer and knowledge acquisition. The same authors argue that cooperating universities contribute to innovation generation. For this same reason, managers of companies should give a preference to the participation in value chains or networks of cooperation that use knowledge intensively. In a convergent way, the results presented here point out that cooperative relations positively influence the pro-eco-innovation orientation.

H5 proposes that lean management principles have a positive influence on the pro-eco-innovation orientation. Analysing the total sample, the adoption of lean management principles has a positive and significant productivity effect (OCAP), especially when a company introduces a process innovation. Already focusing on high-tech subsample, lean has a positive and significant effect in terms of quality (OQUA). Nevertheless, it should be noted that quality (OQUA) does not always have a positive impact but only in terms of process innovation, since in product innovation, the results show a negative effect. Thus, in terms of quality (OQUA), the results follow opposite directions when it comes to different types of innovation (product vs. process). This may be linked to the fact that quality in a high-tech enterprise will soon be required, given the drastic reduction in the product life cycle's duration, often through the market-pull effect. The innovation process itself is often associated with an innovated product or service, with a radical nature, which in turn may be more related to product innovation than to process innovation, which usually denotes a more incremental nature.

Therefore, H5 is not rejected. Nevertheless, the results do not show a positive effect of lean management principles in low-tech companies and, indeed, show a negative effect in relation to quality. The problem of eco-innovation has been increasingly analyzed, based on the resource-based view and project management approaches. On the one hand, this makes it possible to determine the value chain and process design. On the other hand, it facilitates the integration of principles and lean process management practices in the scope of a pro-eco-innovation orientation. For instance, Portillo-Tarragona et al. [18] showed that companies implement eco-innovation solutions based on different certifications (e.g., ISO 14001 and ISO 50001 standards), which refer directly to the quality principle. Pacheco et al. [19] emphasized that eco-innovation in a lean environment represents an opportunity for cost reduction, sustainable growth, and the enhancement of corporate image vis-à-vis customers. Thus, although some principles of the lean paradigm have been addressed in themes related to eco-innovation, so far, now the adoption of lean principles and their influence on the pro-eco-innovation orientation remained unexplored.

## 6. Conclusions

As key results, the importance of the technological factor was found to be essential in stimulating eco-innovation and is a determinant factor common to both high-tech and low-tech companies.

The same applies to the market factor, where the focus is essentially on reducing labour costs per unit produced together with an entry into new markets and/or increased market participation, which highlights the importance of the market as a mechanism of pro-eco-innovation orientation. In the case of high-tech companies, public policies are relevant in influencing product innovation, although that influence is not detected for low-tech ones.

Regarding *lean* management principles, there is a significant and positive influence on eco-innovation, in particular, with productivity (OCAP) showing evidence of its positive significance in process innovation (total sample) and quality (OQUA) also showing mixed results of its positive

significance in terms of process innovation and its significance, although negative, in terms of product innovation (high-tech sample). For the flexibility (OFLEX) principle to be tested in the scope of the adoption of lean management principles, there are no conclusive results as to their impact on the pro-eco-innovation orientation.

In terms of the contributions now made, pioneering the adoption of lean management principles as an internal mechanism for strengthening the pro-eco-innovation orientation is the original contribution of this study, which deserves greater prominence. It must also be stressed that cooperation relationships are also relevant in high- and low-tech firms. High-tech firms have relationships with universities or other higher education institutions, which are important for product innovation, while low-tech firms have cooperation relationships with competitors or other companies in the same sector of activity in other European countries regarding process innovation. Still, on the notable cooperative relations with universities, previous findings on eco-innovation research [13] report that the cooperating universities contributed to the creation of innovation less frequently than the R&D institutes.

Besides the influences described above, employer size has a positive and significant influence on the pro-eco-innovation orientation, agreeing with previous empirical evidence found by Cleff and Rennings [67], Bernauer et al. [50], De Marchi [57], LeBas and Poussing [68], and Trigueiro et al. [64], but the same is not found for the variable of the country in which the group headquarters is located. In turn, for the group variable (GP), which determines the influence of belonging to a group of companies, no evidence is found of its significance, contrasting with the view of LeBas and Poussing [68], according to whom belonging to a group has a positive and significant influence on the probability of innovation, via resource availability, particularly R&D, bearing in mind the specific and complex nature of environmental technologies [95].

Concerning the public policy factor, implicit is the promotion of public policies to provide the country, markets, and companies with a greater innovation capacity, which generates new resources or products that can make the economy more competitive and dynamic based on assumptions of a true circular economy functioning with the pro-eco-innovation orientation.

One of the main limitations is in relation to the lack of information in CIS 2012 for the variables necessary to draw more solid conclusions about the determinant factors of eco-innovation. Another limitation is related to the results of applying the logistic regression model, since the majority of the variables considered in the analysis, associated with each determinant factor, would be expected to present significant results. This was not the case in the group of high-tech companies, and their effect on these firms' eco-innovation tendency was inconclusive. Another limitation is the lack of previous studies addressing the effects that the adoption of the lean paradigm can have on eco-innovation and not showing how this behaves in order to influence the adoption of eco-innovative activities positively or negatively. In the specific case of the robust design, considered as important for the adoption of lean management principles, it was impossible to verify its influence on the pro-eco-innovation orientation, since in the CIS 2010, no related variables were found.

Finally, suggested for future research is study of the effects of lean tools as mechanisms potentially influencing the export orientation of companies with different levels of knowledge intensity. In terms of research guidelines, new research avenues are opening concerning the need for more studies exploring the relationship between lean management and eco-innovation and using a project management approach. Also, in the scope of future research endeavours, it is suggested that the empirical operationalization of the challenging construct of a robust design be deepened in order to analyse the effect of the latter on the pro-eco-innovation orientation.

**Author Contributions:** Conceptualization: J.L. and S.d.B.; methodology: J.L., S.d.B., and S.C.; research: J.L., S.d.B., and S.C.; redaction—original preparation of the draft: J.L., S.d.B., and S.C.; redaction—revision and edition: J.L., S.d.B., and S.C.; viewing: J.L., S.d.B., and S.C.; supervision: J.L., S.d.B., and S.C.

**Funding:** This research was funded by the project EMaDeS—Energy, Material, and Sustainable Development EU/CCDRC/FEDER (Brussels/Coimbra, Central Region, Portugal) 2017 to 2021|Central-01-0145-FEDER-000017.

**Acknowledgments:** The authors acknowledge the highly valuable comments and suggestions provided by the editors and reviewers, which contributed to the improvement in the clarity, focus, contribution, and scientific soundness of the current study. The authors gratefully acknowledge the Instituto Nacional de Estatística, for the access given to CIS 2010.

**Conflicts of Interest:** The authors declare there are no conflicts of interest.

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
