# Peer review of "Eco-Innovation Influencers: Unveiling the Role of Lean Management Principles Adoption"

_sustainability, doi:10.3390/su11082225_

Round 1

Reviewer 1 Report

With considerable interest, I read your manuscript focusing on the influence of lean management in the eco-innovation. I found the idea of bringing the lean management discourse, which takes place largely outside the eco-innovation research community, to the target journal wonderful. During the recent years, sustainability in businesses has been a topic of increasing discussion, and I am certain that the eco-innovation discourse possesses potential for making boundary-crossing contributions in research.

While the manuscript displays some promise, it also suffers from numerous problems that would need to be addressed prior to its potential publication in an academic journal.

The article presents interesting arguments as a whole. However, there are flaws in the theoretical framework and in the presentation and discussion of results. Some methodological gaps and an incomplete description of the method have been detected.

In the following I am providing more detailed suggestions which I hope can support you to improve the paper, reporting them not in order of importance but of appearance.

In the introduction the relevant references in the eco-innovation field are missing. 

Most of the proposed research hypotheses are not a novelty and the literature review is not accurate. A good literature review will make the gaps that you fill obvious. In my opinion the gap has not been clearly described in the introduction. I had trouble linking your hypotheses to any gap that has not previously analysed in the literature. However, they are perhaps quite broad and ambitious to attempt to answer all the hypothesis in one paper

Hypothesis H, H1, H2, H3 and H4 suggest that perhaps you have carried out a variety of approaches to measuring the determinants of eco-innovation. However, in my opinion only the H5 (Lean management practices have a positive influence on pro-eco-innovation orientation) actually is a potential contribution, so it should perhaps be rephrased. The manuscript could be focused only in this hypothesis and to report in deep the novelty of your study (lean management practices and eco-innovation).

As regards the method which is based on the article, my consideration is that it is not adequately explained and it is not appropriately associated with the results.

Most of the variables have been widely studied in the previous literature. The authors must improve the description of the method in order to enhance the robustness of the utilized methodology and the reliability of the obtained results.

After the presentation of results, I would have expected a discussion (with appropriate referencing) concerning how your findings complement earlier research in your target journal. I am not sure that the implications of your research are a novelty for eco-innovation research.

Fourth, the conclusion section makes no effort to tie your results to earlier eco-innovation research presented in your target journal.

AS a general remark, your article remains very descriptive and I am not quite sure what the reader learns from your article.

Based on my previous comments I think that your paper cannot be published in its current form. Once again, thank you for the opportunity to review your work. I personally enjoyed reading your work and I wish you the best of luck in your continued research endeavours.

Author Response

Reviewer 1 (Rev. 1)

Comment No.

Page

No.

Section

Comments Reviewer   1

Amendments

Rev.1_1

 2

1. Introduction

In   the introduction the relevant references in the eco-innovation field are   missing. 

According to the reviewer’ suggestion, which   we acknowledge, new references were introduced in the Introduction’s item:

[3] Aldieri, L.; Vinci, C. . Green Economy and   Sustainable Development: The Economic Impact of Innovation on Employment. Sustainability   2018, 10 (10), 3541, doi:10.3390/su10103541.

[13] Hájek, P.; Stejskal, J. Cooperation and   Knowledge Spillover Effects for Sustainable Business Innovation in the   Chemical Industry. Sustainability 2018,   10, 1064, doi:10.3390/su10041064.

The   following paragraphs were added:

For a full sustainable performance of companies, not only economic   factors are needed, but also stronger actions to promote the creation of jobs   related to environmental activities [3].

It is also worthwhile to mention that the role assumed by managers in   the decision-making process, especially in the construction of collaborative   partnerships open to sustainable innovation, assumes a fundamental importance   [13].

Rev.1_2

2

1. Introduction

Most   of the proposed research hypotheses are not a novelty and the literature   review is not accurate. A good literature review will make the gaps that you   fill obvious. In my opinion the gap has not been clearly described in the   introduction. I had trouble linking your hypotheses to any gap that has not   previously analysed in the literature. However, they are perhaps quite broad   and ambitious to attempt to answer all the hypothesis in one paper.

According to the reviewer's suggestion, which we acknowledge, the gap was identified in the introduction and linked to the research hypotheses. The following sentences were added: 
Despite this, the literature on the determinant factors of eco-innovation still lacks further efforts and under different research lenses [17]. In addition, studies addressing the effects of lean principles on eco-innovation are almost non-existent, with no evidence of how these principles behave, so as to positively or negatively influence the adoption of eco-innovative activities. In this line of reasoning and in order to address the gap identified, this study intends to contribute actively to the advancement of knowledge about the emerging topic related to the determinant factors of eco-innovation (see formulation of the research hypotheses: H1; H2; H3; and H4); and it also intends to analyse the possible effects of adopting lean principles on pro-eco-innovation orientation, in order to improve the understanding of the extent to which the adoption of lean principles can act as a determinant of eco-innovation (as stated in the H5 hypothesis).  

Rev.1_3

12

2.3 Determinant factors of   eco-innovation: research hypotheses and conceptual model

Hypothesis   H, H1, H2, H3 and H4 suggest that perhaps you have carried out a variety of   approaches to measuring the determinants of eco-innovation. However, in my   opinion only the H5 (Lean management practices have a positive influence on   pro-eco-innovation orientation) actually is a potential contribution, so it   should perhaps be rephrased. The manuscript could be focused only in this   hypothesis and to report in deep the novelty of your study (lean management   practices and eco-innovation).

Following the reviewer's comment, which we acknowledge, the approach used to measure the determinant factors of eco-innovation is explained, based on the research hypotheses: H1; H2 H3 H4 and H5; revealing that the main contribution of this article lies in the H5 hypothesis’ formulation and test. The following sentences were added:  

Although the current approach on eco-innovation was founded on a previous tested set of determinant factors [17], grouped by the dimensions: technology (H1); market (H2); public policies (H3); and cooperative relations (H4); it is worth noting that this approach differs from the previous ones, since the first technology dimension represented in H1 is based on metrics of: internal R&D activities; external R&D activities; equipment, software, external knowledge, business process management, labour organization and also sources of information. In relation to the market represented in H2, distinct metrics are used: cost savings; entry into new markets; and market share. When addressing public policies represented in H3, these are measured by the provision of funding lines; and with regard to the cooperation represented in H4, it is considered, above all, the relations established with universities, research structures and competitors. The novelty here lies in the formulation and test of the dimension still unexplored of the lean principles adoption represented in H5, making use of success critical variables, such as quality, productivity and flexibility, used to reinforce the pro-eco innovation orientation.  

Rev.1_4

15

3.2 Variables and model   specification

As   regards the method which is based on the article, my consideration is that it   is not adequately explained and it is not appropriately associated with the   results.

Most   of the variables have been widely studied in the previous literature. The   authors must improve the description of the method in order to enhance the   robustness of the utilized methodology and the reliability of the obtained   results.

According to the reviewer's   suggestion, which we acknowledge, it was attempted to improve the description   of the method used. The following sentences and changes were introduced:

As   already mentioned, in order to estimate the proposed model and empirically   test the research hypotheses and sub-hypotheses for the different   determinants of eco-innovation, the present study adopts the logistic   regression model, due to the need to analyze the statistical relationship of   a binary dependent variable in relation to more than one explanatory   variable, that is, how the independent variables influence companies in the   creation and introduction of new products or processes significantly improved   (e.g. eco-innovation of products or processes). The logistic regression model is present in empirical studies dealing   with the same relationship as this research [96–101] and so it is presented as a viable model to carry   out this study.

For Marôco [102] logistic regression is an econometric method that is   used to model the occurrence, in probabilistic terms, of one of the two   achievements of the dependent variable classes, where the independent   variables can be qualitative or quantitative. This method also allows the   evaluation of the significance of each of the independent variables included   in the model. The logistic regression model is the most usual method [103]   that makes use of the maximum likelihood estimation to evaluate the   probability of categorical association [104], and the results section may be verified   through the value obtained for the likelihood Logarithmic as well as through   the ‘p’ value obtained, for assessing if the model accurately represents the   data.

Rev.1_5

23/24

5. Discussion

After   the presentation of results, I would have expected a discussion (with   appropriate referencing) concerning how your findings complement earlier   research in your target journal. I am not sure that the implications of your   research are a novelty for eco-innovation research.

According to the reviewer's comment,   which we acknowledge, it was attempted to improve the explanation how the   current findings complement earlier research in our target journal. The   following sentence was added:

Conversely,   in the context of internal R&D activities, the previous empirical results obtained in [13], also suggest that the   internal acquisition of R&D activities   lead to increased innovation as well as to sustainable performance.

Rev.1_6

25

6. Conclusions

Fourth,   the conclusion section makes no effort to tie your results to earlier   eco-innovation research presented in your target journal.

According to the   reviewer’ suggestion, which we acknowledge, we tried to link the results to   previous findings on eco-innovation research. Thus, the following sentence   was added:

Still on the cooperative relations notably with universities, previous   findings on eco-innovation research [13], report that the cooperating   universities contributed to the creation of innovation less frequently than   the R&D institutes.

We look forward to hearing from you.

Yours sincerely,

The authors

Reviewer 2 Report

Summary

This paper presents an analysis of the determinants of eco-innovation activities based on a survey of manufacturing companies based in Portugal. An extensive literature review is presented, that forms the basis for a conceptual model. The model considers technology, market, public policy, cooperation relationships and lean management factors as possible determinants of pro-eco-innovation orientation of companies. A logistic regression method is applied with sub-samples for 'low tech' and high-tech companies. A number of statistically significant results are obtained and discussed in relation to the extant literature. 

General comments 

Section 2.1 provides a comprehensive overview of definitions of eco-innovation but what definition is adopted for this work? Some reflection on the variety of definitions mentioned and their differences would add value to this section. For instance, you could provide a summary table showing the various elements/scope of the eco-innovation definitions so that the reader can identify the most commonly covered elements.

The literature review is generally very good with comprehensive references and a logical discussion.

The main contribution of the paper is not clearly defined in the conclusions section but, from the title and abstract, it seems that the authors consider that the inclusion of Lean Management Practices as a possible determinant of eco-innovation as being the main contribution. However, I have some concerns about the validity and robustness of this particular aspect of the work.

The theoretical foundation for the lean management practices variable is the weakest of those presented and is limited to a general introduction to the background of Lean Thinking. 

However, my main concern is that I am not convinced that the three measures (Improve the quality of goods and/or services, Increase the production capacity of goods and/or services, Improve flexibility in producing goods and/or services) form a sufficiently robust link to the Lean Management Practices variable. Whilst I agree that they are linked to Lean practices, I do not feel they are sufficient by themselves to characterise the Lean Management Practices variable. For instance, they could equally be representative of 'Robust design practices' (see for example: http://orbit.dtu.dk/en/publications/robust-design-impact-metrics-measuring-the-effect-of-implementing-and-using-robust-design(3f93c104-9f30-44a6-bb3f-73f307d730bb).html). 

If the inclusion of Lean Management Principles in the conceptual model is the main contribution of the paper then the theoretical foundation and the robustness of the link between the measures selected and the variable must be strengthened. Ideally, this would involve the reanalysis of the survey data including a greater number of measures to more precisely establish the link with the Lean Management Practices variable. Failing this, the authors should provide further justification for why you believe these three measures are sufficient to characterise the Lean Management Practices variable. A more extensive review of the literature related to Lean and the links to sustainability performance should be completed.

The results are generally in line with previous results and the authors do a good job of linking their findings to similar results in the discussion and conclusions section. 

Specific comments

When introducing the hypotheses it would be useful to define the term 'pro-eco-innovation orientation.' (or make this link within section 3.2 where you present the dependent variables used to assess the eco-innovation performance of a company).

How were the high and low tech samples identified?

Some descriptive statistics of the sample would be useful e.g. breakdown of company size and low tech vs high tech sub samples.

It may be interesting to discuss the differences between the process and product innovation results, particularly when the relationships are in opposite directions, such as for quality in high tech companies.     

In the conclusions you state: 'Regarding lean management practices, there is a significant and positive influence on eco-innovation.' This statement does not reflect the mixed results obtained and is potentially misleading. Please revise to more accurately summarise the results obtained. 

Author Response

Reviewer 2 (Rev. 2)

Comment No.

Page

No.

Section

Comments Reviewer   2

Amendments

Rev.2_1

4

2.1 Eco-innovation

Section   2.1 provides a comprehensive overview of definitions of eco-innovation but   what definition is adopted for this work? Some reflection on the variety of   definitions mentioned and their differences would add value to this section.   For instance, you could provide a summary table showing the various   elements/scope of the eco-innovation definitions so that the reader can   identify the most commonly covered elements.

According   to the reviewer’s suggestion, which we   ackowledge, we refer to the definition of   eco-innovation adopted in this study, as well as presenting our definition of   eco-innovation aligned with the adoption of lean principles. Thus, the   following paragraph was added:

In this   study, we adopt the definition of eco-innovation formulated in [28], and   complemented in other studies such as [29], [30] and [17] that converge in   the following conceptualization: eco-innovation is the production,   application or exploitation of goods, service, production process,   organizational structure or management method that has a novelty character   for the company or user, throughout its life cycle, representing the   reduction of environmental risks and pollution, including a reduced negative   impact of resource use, for example, energy, compared to relevant alternative   options. To this definition we couple the adoption of lean principles,   advancing with our own definition thus exposed: eco-innovation is the   production, application or exploitation of goods, a service, production   process, organizational structure or management method that is new to the   company or user, and that on an ongoing basis, we must focus on increasing   efficiency, flexibility and productivity, as well as eliminating more waste   reuse, following a circular sustainability logic.

Rev.2_2

25

6. Conclusions

The   main contribution of the paper is not clearly defined in the conclusions   section but, from the title and abstract, it seems that the authors consider   that the inclusion of Lean Management Practices as a possible determinant of   eco-innovation as being the main contribution. However, I have some concerns   about the validity and robustness of this particular aspect of the work.

According to the reviewer's suggestion, which we acknowledge, the original contribution of this article is now indicated in the findings section. The following sentence was added:  

In terms of the   contributions now made, pioneering the adoption of lean management principles   as an internal mechanism for strengthening pro eco-innovation orientation is   the original contribution of this study, which deserves greater prominence.

Rev.2_3

6

11

12

26

2.2 Lean management principles

2.3 Determinant factors of   eco-innovation: research hypotheses and conceptual model

6.Conclusions

The   theoretical foundation for the lean management practices variable is the   weakest of those presented and is limited to a general introduction to the   background of Lean Thinking. 

However,   my main concern is that I am not convinced that the three measures (Improve   the quality of goods and/or services, Increase the production capacity   of goods and/or services, Improve flexibility in producing goods and/or   services) form a sufficiently robust link to the Lean Management Practices   variable. Whilst I agree that they are linked to Lean practices, I do not   feel they are sufficient by themselves to characterise the Lean Management   Practices variable. For instance, they could equally be representative of   'Robust design practices' (see for   example: http://orbit.dtu.dk/en/publications/robust-design-impact-metrics-measuring-the-effect-of-implementing-and-using-robust-design(3f93c104-9f30-44a6-bb3f-73f307d730bb).html). 

According   to the reviewer’s suggestion, which we   ackowledge, we have sought to strengthen the theoretical basis of the   lean paradigm. The following paragraphs/sentences   were added, along the text:

The reduction or elimination of waste alone is not an easy process and   for this reason, in [44] it is argued that the lean paradigm should follow   the following five principles: i) value; ii) value chain; iii) flow   optimization; iv) implementation of a pull system; and (v) seeks perfection.   Following these principles is not only specifying the value of a particular   product in precise terms and this is really what the customer wants, but also   through the value chain identifies and analyzes the flow of value for each   product in the sense of being able to map activities that do not add value.   Moreover, lean principles suggest that after identifying the value chain and   waste, a continuous flow must be created that is characterized by the ability   to produce only what is needed for the moment. Also regarding the principles   to be followed by lean, only the customer's requests should trigger all the   processes. Thus, organizations cannot produce what they think the customer   will need, but what is actually requested and in the exact quantity and   moment, and also the fact that by encouraging continuous improvement at all   levels of the organization, through the continuous listening of the client and   the speed of responses, it will be possible to operate a continuous   improvement of the organization [47]. The principles to be followed by the   lean paradigm are fundamentally what will allow to operate in the elimination   of waste since this is the essential focus of this paradigm. (page 6)

The lean philosophy and eco-innovation are often seen as compatible due to their combined focus on waste reduction. The removal of non-value-added activities suggested by the lean paradigm can provide substantial energy savings and reduce the environmental impact of production systems by identifying opportunities for the integration of lean efforts and innovations [90]. (page 11)  

Considering that companies   continually strive to optimize their operations in general, including the   product development process, [92] it is argued that robust design can be   considered as an integral part of such product development. Following [92],   the robust design has as a preferential focus the action of designing   products with functional performance insensitive to variation and noise.   Thus, it is recognized the importance of robust design in lean management   principles, however, it still seems challenging to operationalize, in   empirical terms, what will be its influence in the pro-eco-innovation   orientation. (page 12)

In the specific case of   the robust design, considered as important for the adoption of lean   management principles, it was impossible to verify its influence on the   pro-eco-innovation orientation, since in the CIS 2010 no related variables   were found. (page 26)

Also in the scope of   future research endeavours, it is suggested to deepen the empirical   operationalization of the challenging construct of a robust design, in order   to analyse the effect of the latter on the pro-eco-innovation orientation. (page 26)

Rev.2_4

24

5.   Discussion

If   the inclusion of Lean Management Principles in the conceptual model is the   main contribution of the paper then the theoretical foundation and the   robustness of the link between the measures selected and the variable must be   strengthened. Ideally, this would involve the reanalysis of the survey data   including a greater number of measures to more precisely establish the link   with the Lean Management Practices variable. Failing this, the authors should   provide further justification for why you believe these three measures are   sufficient to characterise the Lean Management Practices variable. A more   extensive review of the literature related to Lean and the links to   sustainability performance should be completed.

The   results are generally in line with previous results and the authors do a good   job of linking their findings to similar results in the discussion and   conclusions section. 

This situation is solved according to the answer provided to the comment 3 made by reviewer 2 and also by including, in section 5, after discussing and contrasting the results for hypothesis H5, the following sentence:  

Thus, the variables used here, in a pioneering way, to test the adoption of lean principles, are sufficient to advance the better understanding of the lean paradigm, as a determinant of the pro-eco-innovation orientation.

Rev.2_5

15

3.1 Database and   Sample

When   introducing the hypotheses it would be useful to define the term   'pro-eco-innovation orientation.' (or make this link within section 3.2 where   you present the dependent variables used to assess the eco-innovation   performance of a company).

According to the reviewer's suggestion, which we acknowledge, the   pro-eco-innovation orientation is briefly defined in the following sentence:

Both   dependent variables, which are possible to select in the CIS 2010 database:   ORME and OREI; make it easier to understand what we consider as a pro-eco-innovation   orientation, which is precisely the organizational orientation for the   adoption of new products and new processes that aim at improving efficiency,   flexibility and productivity, following a circular sustainability logic.

Rev.2_6

14

3.1 Database   and Sample

How   were the high and low tech samples identified?

Considering   the question raised by the reviewer, which we acknowledge, the criteria’s   identification process of the sub-samples concerning high-tech and low-tech   companies is now described. A new Table 1 was inserted in the manuscript.

Rev.2_7

14

3.1 Database   and Sample

Some   descriptive statistics of the sample would be useful e.g. breakdown of   company size and low tech vs high tech sub samples.

According to the reviewer's suggestion, which we acknowledge, we made the disaggregation of the company size and sub-samples of high-tech and low-tech companies (see Table 2).  

In order to characterize   our total sample as well as the sub-samples of high-tech and low-tech   companies, we present here its composition with respect to the company size   (Table 2). So the total sample has 334 companies of which 253 companies have   below 50 workers and 81 companies have 250 or more workers, in turn, the   sub-sample of high-tech counts with a total of 95 companies among which 26   have below of 50 workers and 69 companies have 250 or more workers. The   sub-sample of low-tech counts on 239 companies among which 184 has less than   50 workers and 55 companies have 250 and more workers.

Rev.2_8

24

5. Discussion

It   may be interesting to discuss the differences between the process and product   innovation results, particularly when the relationships are in opposite   directions, such as for quality in high tech companies.  

According to the reviewer's suggestion, which we acknowledge, we are now discussing the differences between the results obtained for process innovation and product innovation. The following sentences were added: 
Analysing the total sample, the adoption of lean management principles has a positive and significant productivity effect (OCAP), especially when a company introduces a process innovation. Already focusing on high-tech sub-sample lean has a positive and significant effect in terms of quality (OQUA). It should be noted that quality (OQUA) does not always have a positive impact, but only in terms of process innovation, since in product innovation the results show a negative effect. Thus, in terms of quality (OQUA) the results follow opposite directions when it comes to different types of innovation (product vs. process). This may be linked to the fact that quality in a high-tech enterprise is soon required, given the drastic reduction in the product life cycle’s duration, often through the market-pull effect. The innovation process itself is often associated with an innovated product or service, with a radical nature, which in turn may be more related to product innovation, than to process innovation, which usually denotes a more incremental nature.

Rev.2_9

25

6. Conclusions

In   the conclusions you state: 'Regarding lean management practices, there is a   significant and positive influence on eco-innovation.' This statement does   not reflect the mixed results obtained and is potentially misleading. Please   revise to more accurately summarise the results obtained. 

According   to the reviewer's suggestion, which we acknowledge, the results were revised.   The following sentence was added:

In particular, with productivity (OCAP) showing   evidence of its positive significance in process innovation (total sample)   and quality (OQUA) also showing mixed results of its positive significance in   terms of process innovation and its significance, although negative, in terms   of product innovation (high-tech sample). For flexibility (OFLEX) principle   to be tested in the scope of the adoption of lean management principles,   there are no conclusive results as to their impact on the pro-eco innovation   orientation.

We look forward to hearing from you.

Yours sincerely,

The authors

Reviewer 3 Report

The aim of the analysis is interesting but the current version of the manuscript needs major revisions before potential publication.

First, the aim should be evidenced in the abstract and introduction sections.

Second, the methodological procedure should be explained in a clearer way.

Third, the literature review should be enriched in such a way that the role of diffusion processes for environmental sustainability are also evidenced (Aldieri & Vinci, 2018; Hayek & Stejskal, 2018).

References.

Aldieri, L. & Vinci, C. P. (2018). Green Economy and Sustainable Development: The economic impact of innovation on employment. Sustainability, 10, 3541.

Hayek, P. & Stejskal, J. (2018). R&D cooperation and knowledge spillover effects for sustainable business innovation in the Chemical Industry. Sustainability, 10, 1064.

Author Response

Reviewer 3 (Rev.3)

Comment No.

Page

No.

Section

Comments Reviewer   3

Amendments

Rev.3_1

2/3

Abstract

1.Introduction

First,   the aim should be evidenced in the abstract and introduction sections.

According   to the reviewer’s suggestion, which we   ackowledge, the aim of the analysis was evidenced both in the abstract   and in the Introduction’s item:

Abstract

It aims, in the first instance, to complement the approach on the determinants of eco-innovation in the existent literature, by incorporating the novelty related to the analysis of the effects arising from the adoption of the lean management principles. Specifically, it aims to analyze the effects of the previously referred determinant factors both on the economic performance and on the innovative performance of Portuguese industrial and service companies with different levels of technological intensity (high-tech versus low-tech). 

1. Introduction

With this, the overall objective of this study is, in the first instance, to contribute into the literature of determinants of eco-innovation by unveiling the still unexplored role played by the adoption of lean management principles. In specific terms, it aims to provide an innovative application by assessing the differences between companies with different levels of technological intensity (high-tech versus low-tech).  

Rev.3_2

1

5

Second,   the methodological procedure should be explained in a clearer way.

According   to the reviewer’s suggestion, which we   ackowledge, the explanation about the methodological procedure funded on the   use of the logistic regression method was clarified.

According to the comment 4 of reviewer 1,   this was clarified in page 15, section 3.2.2. Variables and model specification

Rev.3_3

2

9

 2.3 Determinant factors of eco-innovation: research hypotheses and   conceptual model

Third,   the literature review should be enriched in such a way that the role of   diffusion processes for environmental sustainability are also evidenced   (Aldieri & Vinci, 2018; Hayek & Stejskal, 2018).

According   to the reviewer’s suggestion, which we   ackowledge, the folowing references were added to the literature review:

Hayek, P. & Stejskal, J. (2018). R&D cooperation and knowledge   spillover effects for sustainable business innovation in the Chemical   Industry. Sustainability, 10, 1064.

The following sentence was   added:

Also   in relation to R&D activities, namely internal R&D activities, the   results presented in [13], confirm   that when companies acquire knowledge from internal sources, this leads to   increased innovation and sustainable performance.

In   addition, the two references mentioned below, were included in the   Introduction’s item, as suggested by reviewer 1.

Aldieri, L. & Vinci, C. P. (2018). Green Economy and Sustainable Development: The   economic impact of innovation on employment. Sustainability, 10, 3541.

Hayek, P. & Stejskal, J. (2018). R&D cooperation and knowledge   spillover effects for sustainable business innovation in the Chemical   Industry. Sustainability, 10, 1064.

We look forward to hearing from you.

Yours sincerely,

The authors

Round 2

Reviewer 1 Report

The paper improved significantly the author made effort to clearly answer the research question; to add some references and to improve the discussion.

However, the manuscript suffers from some problems that would need to be addressed prior to its potential publication in an academic journal.

In the following I am providing more detailed suggestions which I hope can support you to improve the paper.

I would suggest to change the title. You are not analysing tha Eco-innovation determinants. .

In the introduction some references in the eco-innovation field have been added. However,. In my opinion you would have to explore the eco-innovation literature focused on project management because some lean management principles have been explored in that field. Please, consider to analyse the contribution of Pacheco, Portillo and other authors that are focused on eco-innovation and project management.

Some references are not correctly cited. For instance, in the following sentence:

In this study, we adopt the definition of eco-innovation formulated in [28], and complemented in other studies such as [29], [30] and [17] that converge in the following conceptualization: eco-innovation is the production.

Please, check all the references and cite them in the correct form and introduce the name of the cited authors when is necessary.

After the presentation of results, I would have expected a discussion (with appropriate referencing) concerning how your findings complement earlier research in your target journal.

After the revision, I am more convinced that the implications of your research could be considered as a novelty because the lean management has not been previously addressed in the eco-innovation research

However, the discussion section makes no effort to tie your results to earlier eco-innovation research presented in your target journal.

Please, explain why do you consider that “the variables used here, in a pioneering way, to test the adoption of lean principles, are sufficient to advance the better understanding of the lean paradigm, as a determinant of the pro-eco-innovation orientation”.

As regards the method which is based on the article, my consideration is that it is not adequately explained and it is very descriptive. The obtained results have not discussed in deep and they have not appropriately associated with the results.

Most of the variables have been widely studied in the previous literature. Please, be sure that you are doing a relevant contribution to the literature and explain why your variable are different than the previous studies.

You are using variables that have been previously used by other authors (R&D) and that are available (CIS) for a large number of researchers. I understand that you are offering a first approach to the relation between lean management and eco-innovation but more studies are needed in this field, for instance related to Project management.

I sincerely hope that my somewhat critical feedback helps you in developing your manuscript towards a publication and does not discourage you from continuing your valuable work.

Based on my previous comments I think that your paper cannot be published in its current form. Once again, thank you for the opportunity to review your work.

Author Response

Eco-innovation Influencers: Unveiling the role of lean management practices adoption

Manuscript ID: sustainability-458541_R2

Dear Editor Fuli Cao

Firstly, we would like to acknowledge the editor and all the reviewers’ comments, which we acknowledge and consider them as very constructive and helpful for revising the new revised version of our manuscript.

Secondly, we are very pleased with the possibility to revise and resubmit the paper. Considering the answers to the questions raised by the reviewers, we give an overview of what was changed according to each referee’s proposals and constructive comments/suggestions.

Reviewer 1

Comment No.

Page

No.

Section

Comments Reviewer   1

Amendments

Rev.1_1

Title paper

 I would suggest to change the title. You are   not analysing the Eco-innovation determinants.

Following   the reviewer’s comment, which we acknowledge, we changed the title for:

Eco-innovation Influencers: Unveiling the   role of Lean Management Principles Adoption.

Rev.1_2

2

3

1.Introduction

In   the introduction some references in the eco-innovation field have been added.   However, In my opinion you would have to explore the eco-innovation   literature focused on project management because some lean management   principles have been explored in that field. Please, consider to analyse the   contribution of Pacheco, Portillo and other authors that are focused on   eco-innovation and project management

Following   the reviewer’s comment, which we acknowledge, several new references were   added in the following sentence presented in the Introduction’s item:

Only a few of these principles have been explored in the literature on interrelated topics such as eco-innovation, project management and resource-based view. Portillo-Tarragona et al. [18] show in their analysis that companies implement eco-innovation solutions at different levels of the value chain, designing and adopting advanced environmental management systems funded on the ISO 14001 and ISO 5000 standards. Still in the field of lean Pacheco et al. [19] present a proposal for an integrated model focused on the systematic generation of eco-innovations. 

Rev.1_3

4

2.1   Eco-innovation

Some   references are not correctly cited. For instance, in the following sentence:

In   this study, we adopt the definition of eco-innovation formulated in [28], and   complemented in other studies such as [29], [30] and [17] that converge in   the following conceptualization: eco-innovation is the production.

Please,   check all the references and cite them in the correct form and introduce the   name of the cited authors when is necessary.

Following the reviewer’s comment, which we acknowledge, we checked all the references mentioned in the cited paragraph, as well as along the manuscript. In addition, we introduced the name of the cited authors in the correct format, as suggested. The following sentence was introduced:In this study, we adopt the definition of eco-innovation originally presented by Kemp and Pearson [30] and complemented by Horbach et al. [17] stating that:   eco-innovation is the production…

Rev.1_4

25

26

27

5. Discussion

After   the presentation of results, I would have expected a discussion (with   appropriate referencing) concerning how your findings complement earlier   research in your target journal.

After   the revision, I am more convinced that the implications of your research   could be considered as a novelty because the lean management has not been   previously addressed in the eco-innovation research. However, the discussion   section makes no effort to tie your results to earlier eco-innovation research   presented in your target journal.

Following the reviewer’s comment, which we acknowledge, the discussion   was reinforced using several reference studies on eco-innovation research, in   order to strengthen the novelty of our research results.

The following paragraphs were added:

Pacheco et al. [107] provide a systematic literature   review on eco-innovation determinants in manufacturing SMEs, identifying the   available resources (e.g., people, technology and know-how) as the most   important determinant factor. This is justified by the fact that   eco-innovation requires investment in qualified human capital, as well as the   acquisition of technology or knowledge.

In addition, Triguero et al. [65] show that belonging to a high-tech   sector increases the likelihood of occurring eco-innovation, this remark is   ratified through the results now presented.

Regarding cost reduction, Scarpellini et al. [108] emphasize that   eco-innovation projects, which aim to reduce costs, and therefore driven by   an efficiency logic, tend to be oriented towards process innovation. The   latter are more frequent in the context of companies that are not so labor   intensive.

Still regarding public policies,   Scarpellini et al. [110] point out a positive relationship between public   incentives and eco-innovation, arguing that the former would reduce the risk   associated with investing in eco-innovation projects, contributing to   increased profitability. In the same line of thought, the new empirical   evidence now presented supports the basic idea that public policies have a   positive influence on eco-innovation.

According to Hájek et al. [13] universities should establish   cooperative relationships with companies in compatible research areas, in   order to foster the practical application of both knowledge transfer and   knowledge acquisition. The same authors argue that cooperating universities   contribute to innovation generation. For this same reason, managers of   companies should give preference to participation in value chains or networks   of cooperation that use knowledge intensively. In a convergent way, the   results presented here point out that cooperative relations positively   influence the pro-eco-innovation orientation.

The   problematic of eco-innovation has been increasingly analyzed, based on the   resource-based view and project management approaches. On the one hand, this makes   possible to determine the value chain and process design. On the other hand,   it facilitates the integration of principles and lean process management   practices, in the scope of a pro-eco-innovation orientation. For instance, Portillo-Tarragona   et al. [18] show that companies implement eco-innovation solutions, based on   different certifications (e.g. ISO 14001 and ISO 50001 standards), which   refer directly to the quality principle. For its turn, Pacheco et al. [19] emphasize that   eco-innovation in a lean environment represents an opportunity for cost   reduction, sustainable growth and enhancement of corporate image vis-à-vis   customers. Thus, although some principles of the lean paradigm have been   addressed in themes related to eco-innovation, so far now the adoption of   lean principles and their influence on the pro-eco-innovation orientation   remained unexplored.

Rev.1_5

Please,   explain why do you consider that “the variables used here, in a pioneering   way, to test the adoption of lean principles, are sufficient to advance the   better understanding of the lean paradigm, as a determinant of the   pro-eco-innovation orientation”.

We   acknowledge the reviewer’s comment and we try to clarify our point by   presenting a new diagram (see Figure 1), which links the lean principles and   the variables used for representing the lean management principles adoption,   although we recognize it is an incomplete representation of those referred   principles, in face of the data available through CIS 2010.

Rev.1_6

16/17

3.2 Variables and Specification

As   regards the method which is based on the article, my consideration is that it   is not adequately explained and it is very descriptive. The obtained results   have not discussed in deep and they have not appropriately associated with   the results.

Following the reviewer’s comment, which we   acknowledge, the explanation about the method was revised, by inserting the   new paragraphs presented below:

In other words, logistic   regression is a statistical technique that aims to produce, from a set of   observations, a model that allows the prediction of values taken by a   categorical variable, often binary, from a series of continuous explanatory   variables and/or binary. In the regression analysis logistic model, the   regression model of the probability of observing a given event is expressed   as follows:

                                                                          (1)

In equivalent terms:

                                                                            (2)

Where:

L=

Where: P estimate is the   estimated probability of a given event occurring, and ≈2,718, Nepper number,   is the value used in the exponential function, and, β_1, β_2, ..., β_k and k, are the estimated regression coefficients   that correspond to the k independent variables, with β_0 the estimated model constant. The parameters   (coefficients) considered are estimated by means of the maximum likelihood   method, which consists in determining the values of the parameters that   maximize the probability of obtaining the set of observed values. According   to Hosmer and Lemeshow [103], the maximum likelihood method allows estimating   the regression coefficients, which maximizes the probability of obtaining the   realizations of the dependent variable of the sample. The likelihood function   expresses the probability of observed data, such as unknown parameters. The   maximum likelihood estimators of these parameters, are chosen in order to   obtain the maximum likelihood, being expressed in the following system of   equations:

 ó  , j = 1, ..., m    

Rev.1_7

27

28

6. Conclusions

Most   of the variables have been widely studied in the previous literature. Please,   be sure that you are doing a relevant contribution to the literature and   explain why your variable are different than the previous studies. You are   using variables that have been previously used by other authors (R&D) and   that are available (CIS) for a large number of researchers. I understand that   you are offering a first approach to the relation between lean management and   eco-innovation but more studies are needed in this field, for instance   related to Project management.

Following   the reviewer’s comment, which we acknowledge, a short sentence providing a   guideline for future research, stating that further studies are required on   the relationship between lean management and eco-innovation, using a project   management approach. Thus, at the end of the 6. Conclusions’ item, the   following sentence was added:

In terms of research   guidelines, new research avenues are open concerning the need for more   studies exploring the relationship between lean management and   eco-innovation, using a project management approach.

We look forward to hearing from you.

Yours sincerely,

The authors

Reviewer 2 Report

I would like to thank the authors for taking the time to address the comments provided. Below are some further comments.

Rev.2_1 - The wording of the definition provided would benefit from a little rewording, particular the last part "... and that on an ongoing basis, we must focus on increasing   efficiency, flexibility and productivity, as well as eliminating more waste   reuse, following a circular sustainability logic." Who are the 'we' in this? "...eliminating more waste reuse..." - this does not make sense, are there some words missing? "...following a circular sustainability logic" - you are introducing more terms that need defining in their own right; probably better to leave this last part out.

Rev.2_2 - Amendment is fine.

Rev.2_3 - My apologies if this comment was not clear. I was not expecting you to integrate the robust design reference in the paper. I was instead making the point that the three measures that were selected as being representative of lean management principles could also be representative of other types of management principles. I am questioning the construct validity.

You are claiming that the three measures in the survey (Improve the quality of goods and/or services, Increase the production capacity of goods and/or services, Improve flexibility in producing goods and/or services) are exclusively linked to the principles of lean management. I believe that they are not sufficient by themselves. You have highlighted five lean management principles from reference 44 (Value, Value chain, Flow optimisation, Pull system,Seeks perfection) but I do not see how these strengthen the link with the three measures selected ((Improve the quality of goods and/or services, Increase the production capacity of goods and/or services, Improve flexibility in producing goods and/or services). Perhaps you could include a simple diagram to explain how you envisage the links between the measures and the five lean principles you have identified?

As it currently stands explanation of the link does not convince me as I could make the same link between the three measures and other management principles (such as robust design principles)

Rev.2_4 - As above.

Rev.2_5 - Amendment is fine.

Rev.2_6 - Amendment is fine.

Rev.2_7 - Amendment is fine.

Rev.2_8 - Amendment is fine.

Rev.2_9 - Amendment is fine.

Author Response

Dear Editor Fuli Cao

Firstly, we would like to acknowledge the editor and all the reviewers’ comments, which we acknowledge and consider them as very constructive and helpful for revising the new revised version of our manuscript.

Secondly, we are very pleased with the possibility to revise and resubmit the paper. Considering the answers to the questions raised by the reviewers, we give an overview of what was changed according to each referee’s proposals and constructive comments/suggestions.

Reviewer 2

Comment No.

Page

No.

Section

Comments Reviewer   2

Amendments

Rev.2_1

5

2.1 Eco-innovation

The wording of the definition   provided would benefit from a little rewording, particular the last part   "... and that on an ongoing basis, we   must focus on increasing   efficiency, flexibility and productivity, as   well as eliminating more waste   reuse, following a circular   sustainability logic." Who are the 'we' in this?   "...eliminating more waste reuse..." - this does not make sense,   are there some words missing? "...following a circular sustainability   logic" - you are introducing more terms that need defining in their own   right; probably better to leave this last part out.

Considering the reviewers’ comment, which we   acknowledge, the sentence was revised in the following terms:

…..focus   should be on increasing efficiency, flexibility and productivity, as well as   eliminating more waste, following the logic of circular sustainability.

Rev.2_2

Amendment is fine.

Rev.2_3

6/7

2.2 Lean management principles

You are claiming   that the three measures in the survey (Improve the quality of goods and/or   services, Increase the production capacity of goods and/or   services, Improve flexibility in producing goods and/or services) are   exclusively linked to the principles of lean management. I believe that they   are not sufficient by themselves. You have highlighted five lean management   principles from reference 44 (Value, Value chain, Flow optimisation, Pull   system, Seeks perfection) but I do not see how these strengthen the link with   the three measures selected ((Improve the quality of goods and/or   services, Increase the production capacity of goods and/or   services, Improve flexibility in producing goods and/or services).   Perhaps you could include a simple diagram to explain how you envisage the   links between the measures and the five lean principles you have identified?

As it currently   stands explanation of the link does not convince me as I could make the same   link between the three measures and other management principles (such as   robust design principles)

Following the reviewer’s   comment, a diagram (see Figure 1) on the lean management principles and lean   measures was introduced in the manuscript, as well as the following sentence:

After identifying and briefly characterizing the five basic principles   of lean philosophy, a diagram (see Figure 1) is presented here to provide a   better understanding of the link between the lean principles and the   measurement measures adopted in this study, in order to be able to assess the   influence of the adoption of lean principles in terms of the   pro-eco-innovation orientation.

Figure   1

Thus, as shown in Figure 1, the   adoption of the five lean principles in an organizational context requires   the implementation of management practices that use lean measures,   represented by: quality; productivity; flexibility. Although the latter are   an incomplete representation of these principles in view of the limited   access to more detailed information, in the context of the present study, these   three lean measures are used in a pioneering way to measure and test the   influence of the adoption of lean principles on the pro eco-innovation   orientation of companies with different levels of technological intensity.

Rev.2_4

As above.

Rev.2_5

Amendment is fine.

Rev.2_6

Amendment is fine.

Rev.2_7

Amendment is fine.

Rev.2_8

Amendment is fine.

Rev.2_9

Amendment is fine.

We look forward to hearing from you.

Yours sincerely,

The authors

Reviewer 3 Report

The manuscript has been structurally improved. Now, it can be accepted for publication.

Author Response

Dear Reviewer

Many thanks for your time and positive appreciation.

Kind regards

The authors

Round 3

Reviewer 2 Report

I thank the authors for acting on the suggestion to include a diagram to clarify the link between the measures used and the lean management principles. Unfortunately, I remain unconvinced by strength of the connections. 

In Figure 1, I was expecting to see one-to-one or one-to-many connections between the three measures and the five lean principles. Whilst the three measures and the five principles are shown in Figure 1, the connections between them are not elucidated. Nor are they explained in the accompanying text. 

If you claim that the three measures are representative of lean management principles then it should be possible to make the connection between the measures and the principles.

Author Response

Following the reviewer’s comment, which we acknowledge, we reveal the argumentation associated with the connection between lean principles and measures, by inserting and numbering the connections in Figure 1 and including in the body of text the following explanatory paragraphs:

Given the intensification of competition coupled with the fact that markets have become increasingly global, companies have realized that in order to become more competitive it is no longer enough to improve efficiency within the organization itself, and it is therefore necessary to focus on lean management of the entire value [47]. Therefore, as recommended in Figure 1, the adoption of the five principles of lean management is based on the creation and management of a value chain (I), which incorporates the flows optimization (II), in the sense of being implemented a true pull system (III), tending to perfection (IV), thus guaranteeing the creation of processes that generate value added (V). The management of lean principles requires the operationalization of measures that, in the present research, are represented through: (i) quality (to improve the quality of goods and/or services, by guaranteeing the highest quality of the components of the chain of value (I) and the maximum value added (V)); (ii) productivity (to increase the production capacity of goods and/or services, in order to ensure the maximization of the tradeoff between the results and resources allocated throughout the entire value chain (I) and flow optimization processes (II)); and (iii) flexibility (to improve flexibility in the production of goods and/or services, to ensure a greater capacity of management and response of the organization, in terms of the pull system (III) and the perfection of processes and productions (IV)).

The new approach proposed here is not only a formal mechanism for implementing a lean culture in organizations, but also for strengthening the pro-eco-innovation orientation, which can be influenced by achieving positive outcomes such as increased quality, productivity and flexibility. This perspective also allows us to argue that, despite quality, productivity and flexibility being associated with all lean principles, the selected measures can be linked to different principles, adopting a lean philosophy of maximum value added, based on a dual problem: minimization of waste; and maximization of the triad formed by quality, productivity and flexibility.
